# Expanding the Capabilities of Reinforcement Learning via Text Feedback

Yuda Song [* 1]   Lili Chen [* 1]   Fahim Tajwar [1]   Rémi Munos [2]
Deepak Pathak [1]   J. Andrew Bagnell [1 3]   Aarti Singh [1]   Andrea Zanette [1]

## Abstract

The success of RL for LLM post-training stems from an unreasonably uninformative source: a single bit of information per rollout as binary reward or preference label. At the other extreme, distillation offers dense supervision but requires demonstrations, which are costly and difficult to scale. We study text feedback as an intermediate signal: richer than scalar rewards, yet cheaper than complete demonstrations. Textual feedback is a natural mode of human interaction and is already abundant in many real-world settings, where users, annotators, and automated judges routinely critique LLM outputs. Towards leveraging text feedback at scale, we formalize a multi-turn RL setup, RL from Text Feedback (RLTF), where text feedback is *available during training but not at inference*. Therefore, models must learn to internalize the feedback in order to improve their test-time single-turn performance. To do this, we propose two methods: **Self Distillation** (RLTF-SD), which trains the single-turn policy to match its own feedback-conditioned second-turn generations; and **Feedback Modeling** (RLTF-FM), which predicts the feedback as an auxiliary objective. We provide theoretical analysis on both methods, and empirically evaluate on reasoning puzzles, competition math, and creative writing tasks. Our results show that both methods consistently outperform strong baselines across benchmarks, highlighting the potential of RL with an additional source of rich supervision at scale. .

## 1. Introduction

Reinforcement learning (RL) has become the foundational technique in modern LLM post-training, often delivering large gains in instruction-following, helpfulness, and reasoning quality (Ouyang et al., 2022; Guo et al., 2025a). Yet the standard RL signal in these systems is typically a sparse scalar reward (or one-bit preference label) per rollout. This creates a fundamental tension: RL can be remarkably effective at scale, but the outcome of each individual trajectory contains very little information about what went wrong and how to fix it, making learning extremely inefficient when the base model is unable to solve the task.

At the other extreme, distillation (Buciluǎ et al., 2006; Ba & Caruana, 2014; Hinton et al., 2015) and imitation learning provide information-dense supervision: a single demonstration can convey a full solution, or token-level correction for on-policy imitation learning (Ross et al., 2011; Agarwal et al., 2024; Lu & Lab, 2025). However, distillation is not applicable to training frontier models, and collecting demonstrations from humans is not scalable.

Natural-language text feedback is both a natural mode of human interaction and already abundant in practice. Users routinely critique chatbot outputs; tool-mediated workflows (e.g., code execution, unit tests, compiler errors, symbolic checkers) produce structured traces that describe failures; and more broadly, natural-language feedback is the primary medium through which humans teach and correct one another. Beyond its abundance, text feedback also occupies a favorable middle ground in information density, offering the best of both worlds: it is richer than a scalar reward, yet cheaper than a complete demonstration—feedback can localize an error, name a violated constraint, or suggest a fix.

A natural framework for incorporating feedback is multi-turn interaction: the model generates an attempt, feedback is appended to form an extended prompt, and the model revises. One might apply standard multi-turn RL to this setting, treating the conversation as a sequential decision-making problem and optimizing cumulative reward across turns (Zhou et al., 2024; Shani et al., 2024). However, this creates a fundamental asymmetry: during training, feedback can guide revision, but at test time, feedback is often unavailable—users want good outputs on the first try, not a back-and-forth dialogue. Without feedback, the second turn

---
[*]Equal contribution   [1]Carnegie Mellon University [2]Inria [3]Aurora Innovation. Correspondence to: Yuda Song <yudas@andrew.cmu.edu>.

*Proceedings of the 43$^{rd}$ International Conference on Machine Learning*, Seoul, South Korea. PMLR 306, 2026. Copyright 2026 by the author(s).

**Reinforcement Learning from Text Feedback (RLTF)**

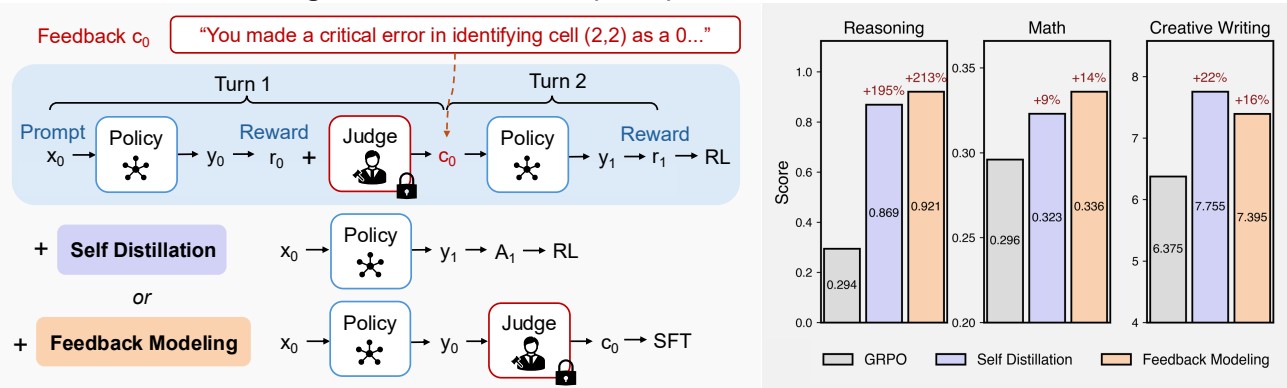

*Figure 1.* Left: performance of our two methods, Self Distillation and Feedback Modeling, on **reasoning puzzles**, **competition math**, and **creative writing** tasks. Both methods outperform standard single-turn GRPO on all three domains. Right: overview of reinforcement learning from text feedback, which uses a feedback provider (judge) to generate critiques $c_0$. Self Distillation trains the policy to match the feedback-conditioned second-turn generations $y_1$, and Feedback Modeling predicts the critiques $c_0$ as an auxiliary objective.

is not even well-defined.[1] With naive multi-turn RL, the policy learns to respond well—leveraging it for *test-time refinement with feedback*—but this does not translate into better first-turn performance when feedback is unavailable. Empirically, we find that naive multi-turn RL improves second-turn performance but yields little gain on the first turn (cf. Section 5). To *internalize* feedback rather than merely condition on it, we need learning objectives that explicitly transform feedback into first-turn supervision. By internalizing text feedback during training, RL can succeed in settings where sparse scalar rewards alone provide insufficient learning signal—effectively expanding the frontier of what reinforcement learning can accomplish.

To address this challenge, we study the setting of RL from Text Feedback (RLTF) and propose two methods to improve first-turn performance from training-time feedback: Self Distillation (RLTF-SD), which treats feedback-conditioned second attempts as implicit demonstrations for the single-turn policy, and Feedback Modeling (RLTF-FM), which learns from feedback itself by predicting critiques as an auxiliary objective. Notably, RLTF-FM can elicit test-time refinement *without* feedback, by rolling out the model's self-critiques during inference. Concretely, our contributions are as follows:

- A formalization of reinforcement learning from text feedback for improving single-turn test-time performance by using feedback during training.

- Two methods to incorporate text feedback into model capabilities: RLTF-SD and RLTF-FM.
- A theoretical justification of our design choices for RLTF-SD, and an extensive theoretical analysis of RLTF-FM via the lens of representation learning.
- Empirical investigation with extensive comparisons and ablations on a suite of diverse benchmarks: Reasoning Gym (Stojanovski et al., 2025), MATH500 (Hendrycks et al., 2021), AIME24, LitBench (Fein et al., 2025) and WritingBench (Wu et al., 2025). Our experiments demonstrate that both of our proposed methods significantly improve single-turn test-time performance over strong baselines that use rewards and text feedback.

## 2. RL from Text Feedback

Let $\mathcal{X}$ be the prompt space and let $\mathcal{X}_0 \subset \mathcal{X}$ be the set of initial prompts that defines the task. Let $\mu \in \Delta(\mathcal{X})$ be the distribution of the initial prompts. We use $\mathcal{Y}$ to denote the output space, and an (LLM) policy $\pi$ maps prompts to distributions over outputs, i.e., $\pi : \mathcal{X} \to \Delta(\mathcal{Y})$. Similarly, let $\mathcal{C}$ be the text feedback space and $\mathcal{M}$ a text feedback provider (human, interpreter, etc.). $\mathcal{M}$ samples text feedback given a prompt and output, i.e., $\mathcal{M} : \mathcal{X} \times \mathcal{Y} \to \Delta(\mathcal{C})$. Finally, let $R : \mathcal{X}_0 \times \mathcal{Y} \to [0, 1]$ be the reward function (that is always evaluated on the original prompt) and $H$ be the horizon of the interaction.

**Interaction protocol.** At the first timestep $h = 0$, a prompt $x_0 \sim \mu(\mathcal{X}_0)$ is sampled, the policy samples an output $y_0 \sim \pi(\cdot \mid x_0)$, receives reward $r_0 = R(x_0, y_0)$, and the feedback provider supplies $c_0 \sim \mathcal{M}(\cdot \mid x_0, y_0)$. For any timestep $h > 0$, the prompt is updated as a function of previous information, $x_h = f(x_{h-1}, y_{h-1}, c_{h-1})$, where $f$ may be as simple as concatenation (e.g., appending $y_{h-1}$

---

[1]This highlights a significant distinction between different modes of text feedback: while some text feedback such as code execution is available during test time (e.g., through tool use), most text feedback such as human feedback is unavailable during test time. In this work we focus on the latter and more challenging setting, and study how to generalize when the feedback is unavailable during test time.

and $c_{h-1}$ to the conversation). The rest follows the first turn: the policy samples $y_h \sim \pi(\cdot \mid x_h)$, obtains reward $r_h = R(x_0, y_h)$, and receives feedback $c_h \sim \mathcal{M}(\cdot \mid x_h, y_h)$. In realistic deployments, $\mathcal{M}$ (or the environment) may terminate the episode early when the output reaches a desired quality (e.g., $r_h = 1$); we will make this explicit whenever early termination is used. We use $\mathbb{P}^\pi$ and $\mathbb{E}^\pi$ to denote the law and expectation over the above interaction process induced by $\pi$, $\mathcal{M}$, and $f$.

**Learning objective.** One natural objective is to maximize the expected sum of rewards over the $H$-turn interaction:

$$J_{\mathsf{MultiTurn}}(\pi) = \mathbb{E}^\pi \left[ \sum_{h=0}^{H-1} r_h \right]. \quad (1)$$

This objective can be optimized using standard multi-turn RL algorithms by treating the interaction as an episodic MDP over the augmented prompts $x_h$ (Zhou et al., 2024; Shani et al., 2024). However, Eq. (1) alone does not isolate the role of text feedback: the policy could maximize reward while treating $c_h$ merely as additional context. Indeed, the objective remains well-defined even if feedback were replaced by uninformative tokens; the policy might simply learn to ignore it. We verify this empirically in Section 5, where naive RL improves multi-turn performance but yields little gain in single-turn competence.

**RL from text feedback (RLTF).** Our goal is to leverage text feedback as a *learning* signal that improves the model's single-turn competence, not merely its ability to improve *test-time refinement with feedback*. To formalize this, we define the single-turn objective:

$$J_{\mathsf{SingleTurn}}(\pi) = \mathbb{E}_{x_0 \sim \mu} \left[ \mathbb{E}_{y \sim \pi(\cdot \mid x_0)} [R(x_0, y)] \right], \quad (2)$$

which evaluates the policy on initial prompts $x_0$ without additional feedback at test time. In RLTF, while optimizing Eq. (1) is straightforward with multi-turn RL algorithms, the central research question is then:

> *Given access to feedback-augmented trajectories $\tau$ during training, how can we design learning objectives and algorithms that improve $J_{\mathsf{SingleTurn}}(\pi)$?*

We address this question with two complementary methods, described in Sections 3 and 4.

## 3. Self Distillation

Text feedback is particularly valuable because it often turns an incorrect first attempt into a correct second attempt: after receiving a critique, the same policy can revise its answer and improve. Our goal is to convert this *feedback-conditioned* competence into improvement on the *single-turn* metric (Eq. (2)), so that the policy performs well even

when feedback is unavailable at test time. We propose to do this via Self Distillation: we treat the policy acting under the post-feedback prompt as an implicit teacher, and distill it into the original one-shot policy. In this sense, distillation "compiles away" the need for feedback by turning test-time refinement into a training signal. This gives us higher-quality trajectories than sampling directly from $\pi(\cdot \mid x_0)$, reducing the exploration burden and turning sparse reward learning into learning from corrected solutions.

Concretely, (focusing on the two-turn case,) for each initial prompt $x_0$ we sample a first-turn output $y_0 \sim \pi(\cdot \mid x_0)$, obtain feedback $c_0$, and form the feedback-augmented prompt $x_1 = f(x_0, y_0, c_0)$. We then sample a revised output $y_1 \sim \pi(\cdot \mid x_1)$ and use $y_1$ to update $\pi(\cdot \mid x_0)$ (not $\pi(\cdot \mid x_1)$), thereby directly targeting single-turn performance. This leads to the following RL-style distillation objective that learns from the $y_1$ distribution:

$$\ell_{\mathsf{distill}}(\pi) = \mathbb{E}_{x_1 \sim \mathbb{P}^\pi, y_1 \sim \pi(\cdot \mid x_1)} \left[ \frac{\pi(y_1 \mid x_0)}{\pi_{\mathrm{ref}}(y_1 \mid x_1)} A(x_0, y_1) \right]. \quad (3)$$

Here $\pi_{\mathrm{ref}}$ denotes a reference distribution used for correction, and $A(x_0, y_1)$ is an estimator of the reward $R(x_0, y_1)$[2], and sg denotes the stop-gradient operator. In the following we will omit the dependency on $x_1 \sim \mathbb{P}^\pi$ when it is clear from context. We introduce Eq. (3) to unify several natural objectives with different instantiations of $\pi_{\mathrm{ref}}$ and $A(\cdot)$.

When we set $\pi_{\mathrm{ref}}(\cdot \mid x_1) = \pi(\cdot \mid x_1)$ (i.e., the data-collection distribution for $y_1$), Eq. (3) recovers an off-policy objective with importance-sampling correction. Moreover, taking $A(y_1) = R(x_0, y_1)$ recovers the original single-turn objective in expectation:

$$\mathbb{E}_{y_1 \sim \pi(\cdot \mid x_1)} \left[ \frac{\pi(y_1 \mid x_0)}{\mathrm{sg}\,\pi(y_1 \mid x_1)} R(x_0, y_1) \right] = J_{\mathsf{SingleTurn}}(\pi). \quad (4)$$

Taking gradient with respect to $\pi$ on Eq. (4) gives an unbiased gradient estimator for $\nabla J_{\mathsf{SingleTurn}}(\pi)$ (under the standard support condition, and note that this in general does not hold for other choices of $\pi_{\mathrm{ref}}$). Thus, we can obtain an (unbiased) gradient for the *single-turn* objective $J_{\mathsf{SingleTurn}}(\pi)$ using samples from the *second-turn* policy, effectively leveraging feedback-conditioned rollouts to improve first-attempt performance. Next, we describe some natural choices of $A(\cdot)$ and $\pi_{\mathrm{ref}}$ and their pitfalls, and our design choices that lead to the Self Distillation algorithm. All derivations and proofs from this section can be found in Appendix C.

**Baselines.** For stability and efficiency of policy gradient algorithms, baseline design (i.e., control variates) is cru-

---

[2]We adopt the notation $A(\cdot)$ because the most common unbiased estimator of the reward is an unbiased estimator of the advantage function.

cial to reduce the variance of the policy gradient estimator (Williams, 1992; Schulman et al., 2015; Guo et al., 2025a; Zeng et al., 2025). Thus it is important to derive the most effective baselines for the distillation objective (Eq. (3)) as it inherits the form of the policy gradient objective.

**Second-turn baseline and gradient-signal collapse.** A natural choice is to use the GRPO-style (Guo et al., 2025a) group-mean baseline computed from second-turn rewards, as this is the standard baseline in multi-turn LLM RL (Team et al., 2025; Tan et al., 2025). Concretely, for each prompt $x_0$, given $\{(y_0^i, y_1^i)\}_{i=1}^N$ where $y_0^i \sim \pi(\cdot \mid x_0)$ and $y_1^i \sim \pi(\cdot \mid x_1^i)$ with $x_1^i = f(x_0, y_0^i, c_0^i)$, the advantage estimator is defined as

$$A_i^{(1)} := R(x_0, y_1^i) - \frac{1}{N} \sum_{j=1}^N R(x_0, y_1^j). \quad (5)$$

In the setting of self-distillation (Eq. (3)), with importance-sampling correction $\pi_{\text{ref}}(\cdot) = \pi(\cdot \mid x_1)$, this yields an unbiased gradient up to a constant multiplicative factor:

$$\mathbb{E}\left[\frac{\pi(y_1^i \mid x_0)}{\pi(y_1^i \mid x_1^i)} A_i^{(1)} \nabla \log \pi(y_1^i \mid x_0)\right] = \frac{N-1}{N} \nabla J(\pi).$$

Note that this bias can be removed by using a leave-one-out baseline $\frac{1}{N-1} \sum_{j \neq i} R(x_0, y_1^j)$ instead of the in-sample mean, but generally this does not matter in practice as the optimizer is agnostic to constant scaling of the gradient.

However, this baseline has a more serious issue: **gradient-signal collapse under second-turn mean baselines**. A second-turn group-mean baseline centers rewards using the same second-turn samples: this can be unbiased in expectation, but it exhibits a *point-wise* degeneracy: whenever the group rewards are (nearly) constant, the centered reward-estimations vanish and the update is exactly (or approximately) zero. This failure mode is not rare in the multi-turn setting. Let $R(x_0, y_1) \in \{0, 1\}$ with second-turn success probability $p_1$ for a fixed prompt $x_0$. Then a second-turn mean baseline yields an exactly zero update whenever the group is constant, which occurs with probability $p_1^N + (1-p_1)^N$. In particular, when feedback makes the second-turn policy highly reliable ($p_1 \to 1$), the probability of a non-zero update scales as $1 - p_1^N \approx N(1-p_1)$, so there is no learning signal for the first turn even though the teacher is consistently correct (at the second turn).

**First-turn baseline.** Baselines computed based on first-turn quantities do not suffer from the above in-sample coupling with $y_1^i$, or the gradient collapse issue. Let

$$b^{(0)} := \frac{1}{N} \sum_{j=1}^N R(x_0, y_0^j), \quad A_i^{(0)} := R(x_0, y_1^i) - b^{(0)}, \quad (6)$$

we have (with $\pi_{\text{ref}}(\cdot) = \pi(\cdot \mid x_1)$) that the baseline term

$b^{(0)}$ is 0 in expectation, and therefore

$$\mathbb{E}\left[\frac{\pi(y_1^i \mid x_0)}{\pi(y_1^i \mid x_1^i)} A_i^{(0)} \nabla \log \pi(y_1^i \mid x_0)\right] = \nabla J_{\text{SingleTurn}}(\pi).$$

In addition, the first-turn baseline $b^{(0)}$ does not normalize by post-feedback rewards. When $p_1$ is high but the first-turn policy is still imperfect ($b^{(0)} < 1$), we have $A_i^{(0)} = R(x_0, y_1^i) - b^{(0)} \neq 0$, so the update remains non-trivial and only vanishes when the student itself is already correct. Note that another natural variant that avoids this issue is taking $A_i = R(x_0, y_1^i) - R(x_0, y_0^i)$, which can be interpreted as the improvement from feedback for each specific trajectory, but potentially result in higher variance than Eq. (6). We defer detailed discussions to Appendices C.2 and C.3.

**Bias-variance tradeoff in importance weighting.** Recall that an unbiased estimator of the single-turn policy gradient can be obtained by importance weighting with $\pi_{\text{ref}} = \pi(\cdot \mid x_1)$), but its variance is controlled by the second moment, an expectation over the directly policy ratios between $\pi(\cdot \mid x_1)$ and $\pi(\cdot \mid x_0)$, instead of the logarithmic of the ratio. Therefore, even moderate distribution shift between first- and second-turn policies can induce heavy-tailed weights. For LLM outputs $y$ (long token sequences), this shift compounds across tokens, making the whole gradient estimation high-variance, which hurts the stability and performance of the training. We provide a rigorous statement of the above intuition and empirical validation in Appendix E.5.

This motivates alternatives to full importance sampling. A standard way is to clip the importance ratio, yielding a CISPO-style objective (Chen et al., 2025):

$$\mathbb{E}_{y_1 \sim \pi(\cdot \mid x_1)}\left[\text{clip}\left[\frac{\pi(y_1 \mid x_0)}{\pi_{\text{ref}}(y_1)}, 1-\varepsilon, 1+\varepsilon\right] A(x_0, y_1)\right].$$

Clipping controls variance by truncating rare but high-magnitude ratios, at the cost of a controlled bias that is governed by $\varepsilon$, an additional hyperparameter to tune.

The other extreme is to discard importance weighting entirely, which introduces higher bias but gives the low-variance objective (here we directly provide the gradient):

$$\nabla \ell_{\text{distill}}^{\text{awr}}(\pi) = \mathbb{E}_{y_1 \sim \pi(\cdot \mid x_1)}[A(y_1) \nabla \log \pi(y_1 \mid x_0)],$$

which resembles advantage-weighted regression (AWR) (Peng et al., 2019; Nair et al., 2020), without the exponential weighting, applied to distillation from feedback-conditioned rollouts.

In the experiment we find that variance dominates bias: setting $\pi_{\text{ref}}(\cdot \mid x_1) = \pi(\cdot \mid x_0)$ (note that this is a special case because $x_0$ is part of $x_1$), which removes the importance weighting, consistently improves stability and final performance compared to using $\pi_{\text{ref}}(\cdot \mid x_1) = \pi(\cdot \mid x_1)$ with

full importance correction or the clipped objective. We provide ablation over all variants in Section 5.2. We therefore view mild bias as benign relative to the variance induced by importance sampling in distillation.

**Remark 3.1.** For clarity, the analysis uses the sequence-level importance weight $W(y_1) := \frac{\pi(y_1|x_0)}{\pi(y_1|x_1)}$. For an autoregressive policy this factorizes exactly as

$$W(y_1) = \prod_{t=1}^{T} r_t, \qquad r_t := \frac{\pi(y_{1,t} \mid x_0, y_{1,<t})}{\pi(y_{1,t} \mid x_1, y_{1,<t})}.$$

Thus, token-level IS simply computes $W$ via per-token ratios and is not an approximation. In contrast, CISPO-style or PPO-style token-level objectives can be viewed as a first-order approximation in the per-token log-ratios $\Delta_t := \log r_t$ via $W = \exp(\sum_t \Delta_t) \approx 1 + \sum_t \Delta_t$ when $\Delta_t$ are small. All experiments in this paper use the token-level surrogate following the standard practice (Sheng et al., 2024).

**Rejection sampling.** Our framework also recovers the commonly used Rejection Sampling (or Supervised Fine-tuning (SFT)) (Scheurer et al., 2023) for distillation: for each $\{x_0, y_0, x_1, y_1\}$, perform a likelihood maximization of $\log(\pi(y_1 \mid x_0))$ if $y_1$ is a better response than $y_0$. This procedure can be recovered by Eq. (3) by taking a binary advantage $A(x_0, y_1) = R(x_0, y_1) \in \{0, 1\}$, and choosing the reference as the same single-turn policy with stop-gradient, i.e., $\pi_{\text{ref}}(\cdot \mid x_1) = \text{sg}[\pi(\cdot \mid x_0)]$. With this choice, we similarly remove the importance weighting. In particular, the induced update direction becomes

$$\nabla \ell_{\text{distill}}^{\text{SFT}}(\pi) = \mathbb{E}_{y_1 \sim \pi(\cdot|x_1)}[R(x_0, y_1) \nabla \log \pi(y_1 \mid x_0)].$$

Therefore, in our setting Rejection Sampling is precisely the special case that distills only the correct second-round generations, without any importance-weight variance from off-policy correction.

While simple, the negative samples (i.e., those with $R(x_0, y_1) = 0$) do not contribute to the learning due to the lack of a baseline. In practice, we indeed observe that Rejection Sampling underperforms methods with baselines, as we will show in Section 5.2.

**Final algorithm.** In conclusion, we adopt **(1)** $\pi_{\text{ref}}(\cdot \mid x_1) = \pi(\cdot \mid x_0)$ for AWR-style RL distillation and **(2)** using first-turn mean reward as baseline (Eq. (6)). We summarize the full RLTF-SD algorithm in Algorithm 1.

# 4. Feedback Modeling

Beyond using feedback-conditioned rollouts to improve the policy, we can also treat the critique itself as a supervision signal and explicitly model the feedback provider. This is appealing because feedback $c_h$ is observed at every turn and

is far richer than a scalar reward: it pinpoints the mistakes, providing dense token-level gradients on failure rollouts. To leverage the dense feedback signal, we propose Feedback Modeling: *training the policy to predict the feedback itself.*

**Feedback-prediction loss.** Recall that at each timestep $h$ the feedback provider samples $c_h \sim \mathcal{M}(\cdot \mid x_h, y_h)$. We define a feedback-prediction distribution:

$$p_\pi(c \mid x, y) := \pi(c \mid f_{\text{FeeMol}}(x, y)),$$

where $f_{\text{FeeMol}}$ is a prompt template that elicits critique-style feedback given $(x, y)$; see examples in Appendix E.1. Using tuples $(x_h, y_h, c_h)$ collected from interaction trajectories, we optimize the cross-entropy objective

$$\ell_{\text{FeeMol}}(\pi) := \mathbb{E}_\pi \left[ \sum_{h=0}^{H-1} -\log p_\pi(c_h \mid x_h, y_h) \right]. \quad (7)$$

Note that we treat $y_h$ as constants (i.e., no gradient) so that $\ell_{\text{FeeMol}}$ is pure supervised learning on the feedback tokens, rather than introducing additional credit assignment through the sampling process.

**Joint objective with RL.** Similar to the self distillation loss, feedback modeling is used as an auxiliary loss in addition to the regular RL objective:

$$\max_\pi \ J_{\text{MultiTurn}}(\pi) - \lambda_{\text{FeeMol}} \ell_{\text{FeeMol}}(\pi), \quad (8)$$

where $J_{\text{MultiTurn}}(\pi)$ is the multi-turn RL objective (Eq. (1)) and $\lambda_{\text{FeeMol}} \geq 0$ controls the strength of the auxiliary feedback loss.

## 4.1. Theoretical analysis

RLTF-FM trains the model to *predict feedback*, not to explicitly output a corrected answer, so its benefit is not obvious a priori. Appendix D provides an early-stage analysis through the lens of *representation learning* in a frozen-rollout regime (a batch RL setting, where data are effectively drawn from a fixed distribution $d_0$) with log-linear (i.e., softmax) policy with learned representation. The central question is: *what representation directions are statistically identifiable from the available training signal under base rollouts?* With the batch setting and log-linear policy, our setting is rather idealized, but the analysis yields useful insights into the benefit of RLTF-FM, which we summarize below.

**Reward-only signal can be both *rare* and *geometrically concentrated*.** With sparse rewards, especially early in training when the base policy performs poorly, only a small fraction of rollouts succeed. Let $\varepsilon_0$ denote this base pass rate. Then the per-sample policy-gradient estimator has low signal-to-noise ratio, and reliably estimating even a single gradient component can require on the order of $1/\varepsilon_0$ rollouts. Beyond this finite-sample bottleneck, we identify a

population-level geometric limitation: even conditioning on success, the reward-only learning signal can concentrate on a small set of representation directions. Equivalently, there can exist a large subspace of directions that are *weakly identified* by reward-only updates under base-policy sampling. In the frozen-rollout regime, we formalize this by defining a *low-signal subspace* $S_{\text{low}}$ from success-conditioned score statistics at initialization, and we show that progress in $S_{\text{low}}$ along the optimization trajectory is controlled by the cumulative magnitude of the success-conditioned score in that subspace. Intuitively, reward-only RL can therefore behave like an effectively low-rank update under base rollouts, making some task-relevant representation directions difficult to learn without auxiliary supervision.

**Feedback modeling supplies a better-conditioned representation signal.** In contrast, natural-language feedback is dense and structured. We show that under the same batch regime, RLTF-FM induces nontrivial movement of the shared representation under a *coverage* assumtion, which is analogous to standard coverage in linear/low-rank MDPs (Jin et al., 2021; Uehara & Sun, 2021) or LLM preference learining (Chang et al., 2024; Song et al., 2024a). This matches the high-level intuition that *reward-only RL may provide a narrow (often nearly rank-1) representation signal under base rollouts, whereas feedback modeling yields a better-conditioned information source that can "fill in" missing representation directions.*

We summarize the main results informally below; formal statements are in Appendix D.

**Proposition 4.1** (Reward-only bottlenecks under base rollouts (informal))**.** *Under the batch regime, and rewards are sparse with base success rate $\varepsilon_0$. Then reward-only learning faces two bottlenecks:*

(i) Rare-event estimation. *Because reward is supported on a low-probability success event, the directional policy-gradient estimator has low signal-to-noise ratio: for any direction,* SNR *scales at most as $\sqrt{\varepsilon_0}$. Consequently, reliably estimating even a single gradient component requires on the order of $1/\varepsilon_0$ rollouts.*

(ii) Weak identifiability of representation directions. *Even conditioning on success, the reward-weighted gradient signal concentrates on a small set of directions in representation space. Equivalently, there can exist a nontrivial* low-signal subspace *of directions that are weakly identified by reward-only updates under base rollouts. In the frozen-rollout regime, reward-only updates can have negligible projection onto these directions over many steps.*

**Proposition 4.2** (Feedback modeling yields a well-conditioned representation signal (informal))**.** *In the same early-stage frozen-rollout regime, feedback modeling (RLTF-FM) provides an additional supervised learning signal on the shared representation. Under mild conditions on the feed-*

back coverage (*Assumption D.3), RLTF-FM is informative in representation directions that are weakly identified by sparse reward under base rollouts. As a result, RLTF-FM can learn representation degrees of freedom that reward-only RL fails to identify early on.*

The analysis explains why RLTF-FM helps without explicitly teaching revision: predicting critiques supplies an additional supervised signal that is informative in representation directions that are weakly identified by sparse reward under base rollouts. In the batch regime, this feedback signal can induce substantial representation learning in a low-signal subspace. From this perspective, RLTF-FM acts like a *representation preconditioner*: it improves the identifiability and conditioning of the representation degrees of freedom that reward-only RL struggles to learn early on. Beyond offering insights into RLTF-FM, our theory provides a framework and techniques with standalone merits and broader applicability.

### 4.2. Test-time scaling via self-feedback

Because $p_\theta(c \mid x, y)$ is produced by the same policy, the model can be run in a "feedback mode" at inference time to generate critiques and perform iterative refinement: sample $y_0 \sim \pi_\theta(\cdot \mid x_0)$, generate $\tilde{c}_0 \sim p_\theta(\cdot \mid x_0, y_0)$, update $x_1 = f(x_0, y_0, \tilde{c}_0)$, and resample $y_1 \sim \pi_\theta(\cdot \mid x_1)$. This enables test-time scaling without requiring a separate learned judge model; the auxiliary training simply makes the policy's self-critique distribution more faithful to the external feedback channel. We further explore this direction in Section 5.4. The complete training and inference pseudocode is in Algorithm 2.

## 5. Experiments

Our experiments evaluate our two proposed methods for RL from text feedback: RLTF-SD, which uses feedback-conditioned rollouts to improve the single-turn policy via RL, and RLTF-FM, which adds an auxiliary objective that predicts critiques. The goal of our experiments is twofold: (i) quantify how much these components improve performance over standard RL baselines, and (ii) isolate the design choices that make them effective in practice. Concretely, we seek to answer the following research questions:

**RQ1:** How well do self distillation and feedback modeling work across a wide range of tasks?

**RQ2:** Which design choices matter for distillation? In particular, does the proposed design choices (use of baseline, advantage weighted regression) consistently outperform their alternatives?

**RQ3:** How much of the gain remains if we remove rich critiques and provide only a correctness-style signal?

**RQ4:** How does feedback modeling enable effective *test-time* scaling by generating multiple rounds of self-

*Table 1.* Comparison of baselines across **reasoning puzzles**, **competition math**, and **creative writing** tasks. We report single-turn accuracy after 2-turn training (i.e., $J_{\text{SingleTurn}}(\pi)$) of the last checkpoint. For the reasoning tasks and LitBench, we report the mean@1 accuracy, judged by either verifiable reward or LLM-as-a-judge. For the math tasks, we report the mean@32 accuracy from the last checkpoint from the training. The parentheses denote the training dataset. For WritingBench, we follow the official protocol with GPT-4.1-mini as the judge. The accuracy in reasoning and math is normalized between 0 and 1, and the score in creative writing is normalized between 1 and 10. Note that RLTF-SD and RLTF-FM consistently outperform all baselines across tasks.

| | Base Model | GRPO Single turn | GRPO Multi turn | Feedback Descent | RLTF-SD | RLTF-FM |
|---|---|---|---|---|---|---|
| **Reasoning** | | | | | | |
| Knights and Knaves | 0.058 | 0.373 | 0.352 | 0.055 | 0.802 | 0.880 |
| Binary Matrix | 0.001 | 0.125 | 0.950 | 0.005 | 0.976 | 0.978 |
| Shortest Path | 0.034 | 0.385 | 0.384 | 0.035 | 0.830 | 0.905 |
| **Math** | | | | | | |
| MATH500 (DAPO) | 0.376 | 0.526 | 0.523 | 0.415 | 0.548 | 0.567 |
| AIME24 (DAPO) | 0.025 | 0.058 | 0.025 | 0.045 | 0.088 | 0.083 |
| MATH500 (DeepMath) | 0.376 | 0.558 | 0.578 | 0.424 | 0.598 | 0.636 |
| AIME24 (DeepMath) | 0.025 | 0.042 | 0.050 | 0.054 | 0.058 | 0.058 |
| **Creative Writing** | | | | | | |
| LitBench | 4.20 | 6.83 | 6.41 | 8.25 | 8.80 | 8.40 |
| WritingBench | 5.71 | 5.92 | 6.29 | 5.30 | 6.71 | 6.39 |

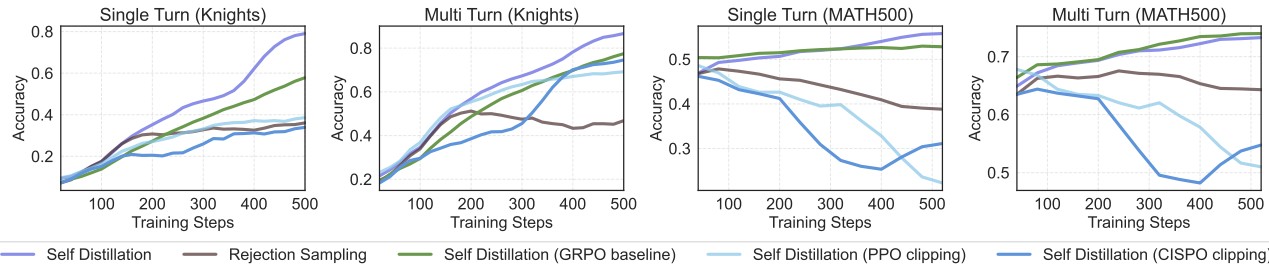

*Figure 2.* Evaluation curves on Knights and Knaves and MATH500 (trained on DAPO) for ablations on self distillation design choices. For each environment: Left: single-turn accuracy; Right: multi-turn accuracy. RLTF-SD (GRPO baseline) denotes using AWR objective with second turn mean baseline. RLTF-SD (PPO clipping) denotes using PPO style clipping on importance weighting with first turn baseline. RLTF-SD (CISPO clipping) denotes using CISPO style clipping on importance weighting with first turn baseline. Note that our proposed design choices consistently outperform the alternatives in both single-turn and multi-turn performance.

feedback at inference time?

**Experiment setup and baselines.** In our experiments, we use the Qwen3-235B-A22B-Instruct-2507 (Yang et al., 2025) model to simulate the feedback provider ($\mathcal{M}$), and we use Llama-3.1-8B-Instruct (Grattafiori et al., 2024) as the learner. We use early termination (c.f. Section 2) unless otherwise specified. We defer the details of each environment and benchmark, as well as prompts for feedback provider and learner to Appendices E.1 and E.3.

We compare with a comprehensive set of baselines: for reward-only RL, we use GRPO (Shao et al., 2024), training with both single-turn ($J_{\text{SingleTurn}}$) and multi-turn ($J_{\text{MultiTurn}}$) objectives; for text feedback aware baselines, we compare with the latest method Feedback Descent (Lee et al., 2025), which performs optimization directly in text space without modifying model weights via pairwise comparisons.

### 5.1. General Results

To investigate **RQ1**, we compare RLTF-SD and RLTF-FM with the baselines across a wide range of tasks, including **reasoning puzzles** (Knights and Knaves, Binary Matrix, Shortest Path) (Stojanovski et al., 2025; Tajwar et al., 2025), **competition math** (training on DAPO (Yu et al., 2025) and Deepmath (He et al., 2025) and testing on MATH500 (Hendrycks et al., 2021) and AIME24), and **creative writing** (LitBench (Fein et al., 2025) and WritingBench (Wu et al., 2025)). We defer the details of benchmarks, prompts, and hyperparameters to Appendices E.1, E.3 and E.4 respectively.

We focus on the 2-turn setting and compare the final single-turn performance $J_{\text{SingleTurn}}(\pi)$. We summarize the main results in Table 1 and we defer the multi-turn performance and evaluation curves to Appendix E.6. We observe that both RLTF-SD and RLTF-FM consistently outperform all

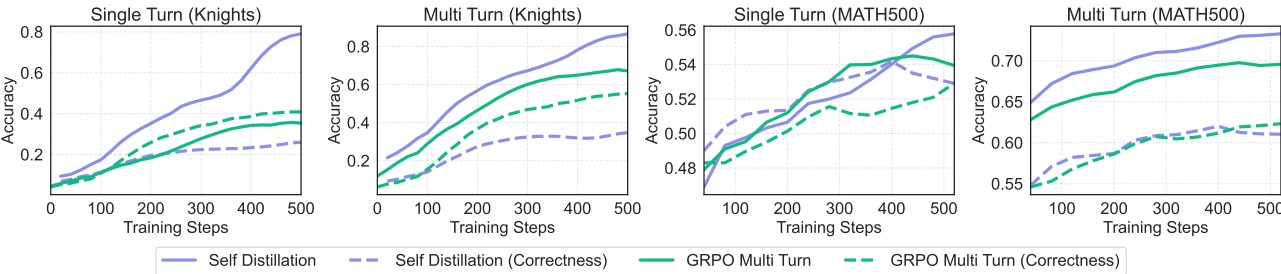

*Figure 3.* Evaluation curves on `Knights and Knaves` and `MATH500` (trained on `DAPO`) for text feedback vs. correctness-only feedback. We compare single- and multi-turn accuracy on two algorithms: multi-turn GRPO and RLTF-SD. Overall, using text feedback outperforms using correctness-only feedback for single-turn and multi-turn accuracy on both algorithms.

baselines across tasks, demonstrating the effectiveness of learning from text feedback. Notably, in terms of single-turn performance GRPO with multi-turn training demonstrates similar performance to single-turn training, suggesting that *naively incorporating feedback as additional context is insufficient to internalize its learning signal.* Feedback Descent also underperforms our methods, indicating the importance of parameter space optimization instead of text space optimization. Although both proposed methods outperform the other baselines across the board, the improvement is more significant in the reasoning tasks and `LitBench`, where the train-test distribution mismatch is small, and thus feedback can significantly accelerate learning. Still, incorporating feedback also helps in domains like math and `WritingBench`[3], indicating the generalization of feedback incorporation. Finally, to compare RLTF-SD and RLTF-FM, we observe that RLTF-SD outperforms in creative writing tasks where the teacher-student distribution mismatch is small, and RLTF-FM obtains better results under math and reasoning tasks where the feedback is more subjective and thus the auxiliary prediction loss is easier to optimize.

**Case studies.** To better understand how text feedback shapes model behavior during training, we qualitatively examine first- and second-turn generations, and show a few examples in Appendix E.2. These examples demonstrate that feedback can help the model escape local optima that RL can get stuck in (e.g., claiming that all problems are infeasible), correct flawed reasoning chains, and identify arithmetic errors. In this way, text feedback provides targeted, actionable information that scalar rewards cannot convey.

### 5.2. Ablation on Self Distillation

In this section, we investigate **RQ2** by performing ablations on the design choices and feedback signal of RLTF-SD. We ablate two major design choices: **(i) the use of a baseline for advantage estimation/variance reduction**, where

---

[3]For `WritingBench` evaluation, we use the same checkpoint from training on the `LitBench` training set, but `WritingBench` has tasks beyond story writing, which is the only task in `LitBench`.

we compare the GRPO style baseline in Eq. (5) and our first-turn baseline in Eq. (6). **(ii) bias-variance tradeoff in importance weighting**, where we compare our AWR-style objective with importance weighting with two different clipping objectives: CISPO style clipping (Chen et al., 2025) and PPO style clipping (Schulman et al., 2017). Finally we also compare with the Rejection Sampling baseline (Scheurer et al., 2023), which adds an SFT auxiliary loss to imitate the correct second-turn responses.

We perform ablation on the `Knights and Knaves` and math reasoning training with `DAPO` and summarize the results in Figure 2. We observe that introducing importance weighting introduces instability during training, even with different mechanisms of clipping on the importance weighting. Without importance weighting, our first-turn baseline provides significant performance improvement over the regular GRPO-style baseline, indicating the empirical benefit of our design beyond the didactic setting considered in Section 3. Notably, Rejection Sampling also underperforms RLTF-SD and RLTF-SD with GRPO baseline, indicating the benefit of variance reduction via baselines.

### 5.3. Ablation on Feedback

To answer **RQ3**, we compare to a correctness-only version of RLTF-SD that does not use text feedback and only provides a correctness signal. Specifically, we replace the judge critique after the first turn with simply the sentence `"Your previous answer was {correct/incorrect}"`. Figure 3 shows the performance of using correctness-only feedback on two algorithms: 1) multi-turn GRPO, and 2) RLTF-SD. We find that the correctness-only baseline does not perform well compared to RLTF-SD, indicating that semantically rich text feedback is critical. One notable exception is the single-turn `Knights and Knaves` accuracy using multi-turn GRPO. Without distillation, neither text feedback nor correctness-only feedback can significantly influence the model's first-turn response, so there is little difference between the two in this setting.

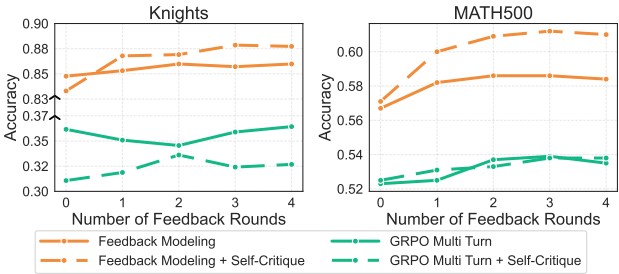

*Figure 4.* Test-time scaling results on `Knights and Knaves` and `MATH500` (trained on `DAPO`). We allow the model to generate multiple rounds of self-feedback at inference time (denoted in the x-axis). We compare `RLTF-FM` with multi-turn scalar-based RL, and the dashed line ("+ Self-Critique") denotes further using RL to improve the self-critique during training (Algorithm 4). We use skipped y-axis for the plot on the left for ease of presentation.

## 5.4. Test-time Scaling of Feedback Modeling

Finally, for **RQ4**, we investigate the test-time scaling ability of RLTF-FM by generating multiple rounds of self-feedback at inference time. Specifically, we evaluate the model trained with `RLTF-FM` on `Knights and Knaves` and `MATH500` (trained with `DAPO`) by allowing it to generate up to 5 rounds of generation with self-feedback at inference time. We introduce an additional baseline where we use RL to improve model's self-critique using second-turn reward (Algorithm 4), and we disable early termination during the training under this setting.

We summarize the results in Figure 4. We make the following observations: first, using RL for learning self-critique is not sufficient: in the math experiment, we observe that GRPO with and without self-critique training achieve similar test-time improvement. Second, adding RLTF-FM loss in addition to self-critique RL training brings significant test-time improvement. Third, the benefit of RLTF-FM is mainly in terms of the magnitude of improvement, not in terms of the number of rounds of improvement. The test-time improvement saturates after a handful of rounds, but this is expected and corroborates with the self-improvement literature (Huang et al., 2023; Song et al., 2024b).

## 6. Related Work

**Learning from text feedback.** A well-studied area of human-robot interaction explores learning from natural language corrections (Broad et al., 2017; Sharma et al., 2022; Bucker et al., 2022; Liu et al., 2023a; Cui et al., 2023; Lynch et al., 2023; Liang et al., 2024; Shi et al., 2024). In these approaches, humans provide corrections such as "move a bit to the left," grounded in the robot's perception and action space and used to update policies or value functions. Zhao et al. (2026) incorporates text feedback for image genera-

tion by prompting VLMs to provide critiques of generated images. We study learning from text feedback in the context of RL for LLM reasoning (Shao et al., 2024; Guo et al., 2025a; Hu et al., 2025), which typically relies on a single scalar reward. In contrast, learning directly from text feedback preserves semantic structure and compositionality. The theoretical benefit of learning from text feedback has been shown in Pukdee et al. (2023); Xu et al. (2025). Feng et al. (2024); Hong et al. (2025); Zhang et al. (2025c); Yang et al. (2026) study how to incorporate self-critiques into the model (e.g., via policy and value distillation), but the setting differs in that they do not assume access to external text feedback. Another class of methods (Chang et al., 2023; Amani et al., 2025; Li et al., 2025; Zhang et al., 2025b) bridges SFT and RL by revealing partial prefixes of an expert solution to guide RL training. Wang et al. (2025b) converts text feedback to denser span-level rewards, but this ultimately collapses the text into the same order of numerical signals as regular RL. Furthermore, several works (Madaan et al., 2023; Cheng et al., 2024; Yuksekgonul et al., 2024; Lee et al., 2025) have also proposed learning from text feedback by propagating minimal subgraphs, or performing optimization in text space. Finally, in the spirit of goal relabeling (Andrychowicz et al., 2017), feedback-conditioned policies (Liu et al., 2023b; Zhang et al., 2023; Luo et al., 2025) use feedback as a goal in hindsight rather than an intermediate step in a multi-turn interaction.

In Appendix A, we provide additional related work on **LLM distillation**, **World modeling** and **Multi-Turn RL**.

## 7. Conclusion and Discussion

We study RL from text feedback, addressing the sparsity of scalar rewards while providing a scalable alternative to expert demonstration. Our two methods, Self Distillation (RLTF-SD) and Feedback Modeling (RLTF-FM), enjoy favorable theoretical properties and demonstrate strong empirical performance across reasoning, math, and creative writing tasks. As text feedback becomes increasingly abundant through human-AI interaction, we see RL from text feedback as a natural next step beyond reward optimization.

Several limitations suggest directions for future work. First, real-world feedback may be noisy or subjective, likely requiring data curation and filtering. Second, while our methods generalize to arbitrary horizons, truly long-horizon feedback interaction may require techniques such as summarization to address distribution shift and context limits. Third, our theory focuses on representation learning near the base policy's distribution; a fully end-to-end analysis would strengthen understanding of feedback modeling. Finally, exploring interplay with other fine-grained supervision methods, such as process reward models (Lightman et al., 2023) is a promising direction.

## Acknowledgments

The authors are grateful to Thinking Machines for their generous support of this research through the Tinker Research Grant. The authors are are grateful to Daman Arora, Clare Birch, Yoonho Lee, Bingbin Liu, Samuel Sokota, Wen Sun, Cyril Zhang and Yifei Zhou for their insightful discussion and support. AS and YS acknowledge and thank the support of NSF AI Institute for Societal Decision Making AI-SDM grant IIS2229881 and Simons Foundation grant 888970. YS thanks the support of the Two Sigma Fellowship. LC is supported by the NDSEG Fellowship.

## Impact Statement

Our work aims to make language model training more efficient by leveraging natural-language feedback. This could lower the cost of aligning models to user intent and reduce reliance on expensive expert demonstrations. However, the same techniques could in principle be used to train models toward undesirable behaviors if the feedback signal is adversarially constructed or systematically biased. We do not believe these risks are unique to our method—they apply broadly to any approach that learns from human or automated feedback—but they merit ongoing attention as text feedback becomes a more prominent training signal.

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

# A. Additional Related Work

**LLM distillation.** In knowledge distillation (Hinton et al., 2015; Kim & Rush, 2016; Sanh et al., 2019), a student model aims to mimic a teacher model's soft probability distribution. On-policy distillation (Agarwal et al., 2024; Xu et al., 2024; Gu et al., 2023; Lu & Lab, 2025; Xiao et al., 2026; Yang et al., 2025) trains the student on its own generations instead of the teacher's generations. In self-distillation (Askell et al., 2021; Snell et al., 2022; Choi et al., 2022; Kujanpää et al., 2025; Mitra & Ulukus, 2025), a student model learns from a teacher that has access to privileged information through its prompt. The teacher and student are typically the same base model; the teacher is not inherently more capable, but instead benefits from additional context embedded in the prompt. Prior work has explored self-distillation across a range of applications, including alignment (Askell et al., 2021), instruction following (Snell et al., 2022), and persona-conditioned dialogue (Choi et al., 2022). Kujanpää et al. (2025) studies how models can learn from unstructured, free-form documents via prompt distillation. Mitra & Ulukus (2025) apply self-distillation to reasoning tasks, via a teacher model with access to both correct and incorrect solutions.

**LLM world models.** World modeling (Sutton, 1991; Ha & Schmidhuber, 2018; Hafner et al., 2020a;b; 2023) has long been used to improve the sample efficiency of RL. An agent learns to predict future states and rewards given the current state and action, and this internal model enables planning through imagined rollouts rather than direct interaction with the environment. More recently, this idea has been adapted to LLMs (Gu et al., 2024; Guo et al., 2025b; Chae et al., 2024; Hao et al., 2023). In this direction, Zhang et al. (2025a) proposed to have LLMs learn from their own collected interaction data ("early experience"), via an implicit world modeling strategy, which uses next-state prediction to learn the environment dynamics. Copet et al. (2025) released Code World Model (CWM), a 32-billion-parameter LLM trained on large amounts of state-action pairs of Python interpreter traces and interactions with Docker environments.

**Multi-turn RL.** In the context of LLMs, generating a complete response and receiving a reward signal without intermediate intervention is often sufficient as there is no need for interaction. However, as LLMs are increasingly deployed as autonomous agents, the need for multi-turn RL has grown significantly. Recently, multi-turn RL (Zhou et al., 2024; Kumar et al., 2024; Abdulhai et al., 2023) has been studied more extensively for agentic settings where interacting with an external environment is beneficial, such as interacting with the terminal (Liu et al., 2023c) or the Internet (Zhou et al., 2023). Several methods (Wang et al., 2025a; Ji et al., 2024; Zhou et al., 2025) have been developed to improve sample complexity and long-horizon performance for multi-turn RL. In our work, this "environment" is the feedback provider, which impacts the model's second-turn generation by critiquing its first-turn response.

# B. Omitted Algorithms

---

**Algorithm 1** Self Distillation

---

**require** Initial policy $\pi_\theta$; group size $N$; learning rate $\eta$; steps $T$, optimizer OPT.

1: **for** $t = 1, 2, \ldots, T$ **do**
2:     Sample a minibatch of prompts $\{x_0^b\}_{b=1}^B \sim \rho$
3:     **for** $b = 1, 2, \ldots, B$ **do**
4:         **for** $i = 1, 2, \ldots, N$ **do**
5:             Sample first-turn output $y_0^{i,b} \sim \pi_\theta(\cdot \mid x_0^b)$
6:             Obtain feedback $c_0^{i,b} \sim \mathcal{M}(x_0^b, y_0^{i,b})$
7:             Form second-turn state $x_1^{i,b} \leftarrow f(x_0^b, y_0^{i,b}, c_0^{i,b})$
8:             Sample second-turn output $y_1^{i,b} \sim \pi_\theta(\cdot \mid x_1^{i,b})$
9:             Get rewards $r_0^{i,b} \leftarrow R(x_0^b, y_0^{i,b})$ and $r_1^{i,b} \leftarrow R(x_0^b, y_1^{i,b})$
10:            Compute return $R^{i,b} \leftarrow r_0^{i,b} + \gamma r_1^{i,b}$
11:         Compute baselines $b^{(0)} \leftarrow \frac{1}{N} \sum_{i=1}^N r_0^{i,b}$, $b^{(R)} \leftarrow \frac{1}{N} \sum_{i=1}^N R^{i,b}$, and $b^{(1)} \leftarrow \frac{1}{N} \sum_{i=1}^N r_1^{i,b}$
12:         Compute self distillation advantages $A^{i,b} \leftarrow r_1^{i,b} - b^{(0)}$ for all $i \in [N]$
13:         Compute first turn RL advantages $A_{\mathsf{RL},0}^{i,b} \leftarrow R^{i,b} - b^{(R)}$ for all $i \in [N]$
14:         Compute second turn RL advantages $A_{\mathsf{RL},1}^{i,b} \leftarrow r_1^{i,b} - b^{(1)}$ for all $i \in [N]$
15:         Form self distillation gradient estimate

$$\widehat{g}^b \;\leftarrow\; \frac{1}{N} \sum_{i=1}^N A^{i,b} \, \nabla_\theta \log \pi_\theta(y_1^{i,b} \mid x_0^b)$$

16:         Form RL gradient estimate

$$\widehat{g}_{\mathsf{RL}}^b \;\leftarrow\; \frac{1}{N} \sum_{i=1}^N \Big[ A_{\mathsf{RL},0}^{i,b} \, \nabla_\theta \log \pi_\theta(y_0^{i,b} \mid x_0^b) + A_{\mathsf{RL},1}^{i,b} \, \nabla_\theta \log \pi_\theta(y_1^{i,b} \mid x_1^{i,b}) \Big]$$

17:         Update policy: $\theta \leftarrow \mathrm{OPT}(\theta, \eta, \widehat{g}^b + \widehat{g}_{\mathsf{RL}}^b)$
18: **return** $\pi_\theta$

---

**Algorithm 2** Feedback Modeling with Test-time Self-Feedback

---

**require** Initial policy $\pi_\theta$; number of self-critique steps $H$.

1: Sample prompt $x_0 \sim \rho$
2: **for** $h = 1, 2, \ldots, H$ **do**
3:     Sample output $y_h \sim \pi_\theta(\cdot \mid x_{h-1})$
4:     Generate self-critique $\tilde{c}_h \sim p_\theta(\cdot \mid x_{h-1}, y_h)$
5:     Form next state $x_h \leftarrow f(x_{h-1}, y_h, \tilde{c}_h)$
6: **return** final output $y_H$

---

---

**Algorithm 3** Feedback Modeling

---

**require** Initial policy $\pi_\theta$; group size $N$; learning rate $\eta$; steps $T$, optimizer OPT.

1: **for** $t = 1, 2, \ldots, T$ **do**
2:     Sample a minibatch of prompts $\{x_0^b\}_{b=1}^B \sim \rho$
3:     **for** $b = 1, 2, \ldots, B$ **do**
4:         **for** $i = 1, 2, \ldots, N$ **do**
5:             Sample first-turn output $y_0^{i,b} \sim \pi_\theta(\cdot \mid x_0^b)$
6:             Obtain feedback $c_0^{i,b} \sim \mathcal{M}(x_0^b, y_0^{i,b})$
7:             Form second-turn state $x_1^{i,b} \leftarrow f(x_0^b, y_0^{i,b}, c_0^{i,b})$
8:             Sample second-turn output $y_1^{i,b} \sim \pi_\theta(\cdot \mid x_1^{i,b})$
9:             Get rewards $r_0^{i,b} \leftarrow R(x_0^b, y_0^{i,b})$ and $r_1^{i,b} \leftarrow R(x_0^b, y_1^{i,b})$
10:            Compute return $R^{i,b} \leftarrow r_0^{i,b} + \gamma r_1^{i,b}$
11:        Compute baselines $b^{(R)} \leftarrow \frac{1}{N}\sum_{i=1}^N R^{i,b}$, and $b^{(1)} \leftarrow \frac{1}{N}\sum_{i=1}^N r_1^{i,b}$
12:        Compute first turn RL advantages $A_{\mathsf{RL},0}^{i,b} \leftarrow R^{i,b} - b^{(R)}$ for all $i \in [N]$
13:        Compute second turn RL advantages $A_{\mathsf{RL},1}^{i,b} \leftarrow r_1^{i,b} - b^{(1)}$ for all $i \in [N]$
14:        Form feedback modeling gradient estimate

$$\widehat{g}^b \ \leftarrow \ \frac{1}{N}\sum_{i=1}^N \nabla_\theta \log \pi_\theta\Big(c_0^{i,b} \mid f_{\mathsf{FeeMol}}(x_0^b, y_0^{i,b})\Big)$$

15:        Form RL gradient estimate

$$\widehat{g}_{\mathsf{RL}}^b \ \leftarrow \ \frac{1}{N}\sum_{i=1}^N \Big[ A_{\mathsf{RL},0}^{i,b} \, \nabla_\theta \log \pi_\theta(y_0^{i,b} \mid x_0^b) + A_{\mathsf{RL},1}^{i,b} \, \nabla_\theta \log \pi_\theta(y_1^{i,b} \mid x_1^{i,b}) \Big]$$

16:        Update policy: $\theta \leftarrow \mathrm{OPT}(\theta, \eta, \widehat{g}^b + \widehat{g}_{\mathsf{RL}}^b)$
17: **return** $\pi_\theta$

---

**Algorithm 4** Feedback Modeling with Self-Critique

---

**require** Initial policy $\pi_\theta$; group size $N$; learning rate $\eta$; steps $T$, optimizer OPT.

1: **for** $t = 1, 2, \ldots, T$ **do**
2:     Sample a minibatch of prompts $\{x_0^b\}_{b=1}^B \sim \rho$
3:     **for** $b = 1, 2, \ldots, B$ **do**
4:         **for** $i = 1, 2, \ldots, N$ **do**
5:             Sample first-turn output $y_0^{i,b} \sim \pi_\theta(\cdot \mid x_0^b)$
6:             Obtain feedback $c_0^{i,b} \sim \mathcal{M}(x_0^b, y_0^{i,b})$
7:             Sample self-critique $\tilde{c}_0^{i,b} \sim p_\theta(\cdot \mid f_{\mathsf{FeeMol}}(x_0^b, y_0^{i,b}))$
8:             Form second-turn state $x_1^{i,b} \leftarrow f(x_0^b, y_0^{i,b}, c_0^{i,b}), \tilde{x}_1^{i,b} \leftarrow f(x_0^b, y_0^{i,b}, \tilde{c}_0^{i,b})$
9:             Sample second-turn output $y_1^{i,b} \sim \pi_\theta(\cdot \mid x_1^{i,b}), \tilde{y}_1^{i,b} \sim \pi_\theta(\cdot \mid \tilde{x}_1^{i,b})$
10:            Get rewards $r_0^{i,b} \leftarrow R(x_0^b, y_0^{i,b}), r_1^{i,b} \leftarrow R(x_0^b, y_1^{i,b})$ and $\tilde{r}_1^{i,b} \leftarrow R(x_0^b, \tilde{y}_1^{i,b})$
11:            Compute return $R^{i,b} \leftarrow r_0^{i,b} + \frac{\gamma}{2}\left(r_1^{i,b} + \tilde{r}_1^{i,b}\right)$
12:         Compute baselines $b^{(R)} \leftarrow \frac{1}{N}\sum_{i=1}^N R^{i,b}$, and $b^{(1)} \leftarrow \frac{1}{N}\sum_{i=1}^N r_1^{i,b}, \tilde{b}^{(1)} \leftarrow \frac{1}{N}\sum_{i=1}^N \tilde{r}_1^{i,b}$
13:         Compute first turn RL advantages $A_{\mathsf{RL},0}^{i,b} \leftarrow R^{i,b} - b^{(R)}$ for all $i \in [N]$
14:         Compute second turn RL advantages $A_{\mathsf{RL},1}^{i,b} \leftarrow r_1^{i,b} - b^{(1)}, \tilde{A}_{\mathsf{RL},1}^{i,b} \leftarrow \tilde{r}_1^{i,b} - \tilde{b}^{(1)}$ for all $i \in [N]$
15:         Form feedback modeling gradient estimate

$$\widehat{g}^b \leftarrow \frac{1}{N}\sum_{i=1}^N \nabla_\theta \log \pi_\theta\left(c_0^{i,b} \mid f_{\mathsf{FeeMol}}(x_0^b, y_0^{i,b})\right)$$

16:         Form RL gradient estimate

$$\widehat{g}_{\mathsf{RL}}^b \leftarrow \frac{1}{N}\sum_{i=1}^N \left[A_{\mathsf{RL},0}^{i,b} \nabla_\theta \log \pi_\theta(y_0^{i,b} \mid x_0^b) + A_{\mathsf{RL},1}^{i,b} \nabla_\theta \log \pi_\theta(y_1^{i,b} \mid x_1^{i,b}) + \tilde{A}_{\mathsf{RL},1}^{i,b} \nabla_\theta \log \pi_\theta(\tilde{c}^{i,b} \mid f_{\mathsf{FeeMol}}(x_0^b, y_0^{i,b}))\right]$$

17:         Update policy: $\theta \leftarrow \mathrm{OPT}(\theta, \eta, \widehat{g}^b + \widehat{g}_{\mathsf{RL}}^b)$
18: **return** $\pi_\theta$

---

# C. Theory Results from Section 3

## C.1. Properties of Baselines

**Setup and notation.** Fix a prompt $x_0$. For $i = 1, \ldots, N$, we sample

$$y_0^i \sim \pi(\cdot \mid x_0), \qquad x_1^i = f(x_0, y_0^i, c_0^i), \qquad y_1^i \sim \pi(\cdot \mid x_1^i),$$

and define rewards $r_0^i := r(x_0, y_0^i)$ and $r_1^i := r(x_0, y_1^i)$. We consider the importance-corrected score-function estimator for the single-turn objective

$$J(\pi) = \mathbb{E}_{x_0 \sim \rho}\big[\mathbb{E}_{y \sim \pi(\cdot \mid x_0)}[r(x_0, y)]\big].$$

For a fixed $x_0$, define

$$g_i := \frac{\pi(y_1^i \mid x_0)}{\pi(y_1^i \mid x_1^i)} \, r_1^i \, \nabla \log \pi(y_1^i \mid x_0).$$

Under the standard support condition, the expectation of $g_i$ equals the true single-turn policy gradient at $x_0$,

$$\mathbb{E}[g_i \mid x_0] \;=\; \nabla \, \mathbb{E}_{y \sim \pi(\cdot \mid x_0)}[r(x_0, y)].$$

**Proposition C.1** (In-sample second-turn group-mean baseline yields $(1 - \frac{1}{N})$ shrinkage)**.** *For a fixed $x_0$, define the in-sample second-turn mean baseline*

$$\bar{r}_1 \;:=\; \frac{1}{N} \sum_{j=1}^{N} r_1^j, \qquad A_i^{(1)} \;:=\; r_1^i - \bar{r}_1.$$

*Consider the importance-corrected gradient estimator*

$$\hat{G}^{(2)} \;:=\; \frac{1}{N} \sum_{i=1}^{N} \frac{\pi(y_1^i \mid x_0)}{\pi(y_1^i \mid x_1^i)} \, A_i^{(1)} \, \nabla \log \pi(y_1^i \mid x_0).$$

*Then, conditioning on $x_0$,*

$$\mathbb{E}\Big[\hat{G}^{(2)} \mid x_0\Big] \;=\; \Big[1 - \frac{1}{N}\Big] \nabla \, \mathbb{E}_{y \sim \pi(\cdot \mid x_0)}[r(x_0, y)].$$

*Equivalently, the in-sample second-turn mean baseline introduces a multiplicative shrinkage factor $(1 - \frac{1}{N})$ in expectation.*

**Proof.** Fix $x_0$. By exchangeability it suffices to analyze a single index $i$ and then take the average. Write

$$A_i^{(1)} \;=\; r_1^i - \frac{1}{N} \sum_{j=1}^{N} r_1^j \;=\; \Big[1 - \frac{1}{N}\Big] r_1^i - \frac{1}{N} \sum_{j \neq i} r_1^j.$$

Hence

$$\mathbb{E}\left[ \frac{\pi(y_1^i \mid x_0)}{\pi(y_1^i \mid x_1^i)} A_i^{(1)} \nabla \log \pi(y_1^i \mid x_0) \mid x_0 \right] = \Big[1 - \frac{1}{N}\Big] \mathbb{E}\left[ \frac{\pi(y_1^i \mid x_0)}{\pi(y_1^i \mid x_1^i)} r_1^i \nabla \log \pi(y_1^i \mid x_0) \mid x_0 \right]$$
$$- \frac{1}{N} \sum_{j \neq i} \mathbb{E}\left[ r_1^j \cdot \frac{\pi(y_1^i \mid x_0)}{\pi(y_1^i \mid x_1^i)} \nabla \log \pi(y_1^i \mid x_0) \mid x_0 \right].$$

For $j \neq i$, the random variable $r_1^j$ depends only on $(y_0^j, x_1^j, y_1^j)$ and is independent of $(y_0^i, x_1^i, y_1^i)$ conditioned on $x_0$ (since the $N$ rollouts are i.i.d. given $x_0$). Therefore the cross term factors:

$$\mathbb{E}\left[ r_1^j \cdot \frac{\pi(y_1^i \mid x_0)}{\pi(y_1^i \mid x_1^i)} \nabla \log \pi(y_1^i \mid x_0) \mid x_0 \right] = \mathbb{E}[r_1^j \mid x_0] \, \mathbb{E}\left[ \frac{\pi(y_1^i \mid x_0)}{\pi(y_1^i \mid x_1^i)} \nabla \log \pi(y_1^i \mid x_0) \mid x_0 \right].$$

Next, for the second factor, note that

$$\mathbb{E}\left[ \frac{\pi(y_1^i \mid x_0)}{\pi(y_1^i \mid x_1^i)} \nabla \log \pi(y_1^i \mid x_0) \mid x_0 \right] = \mathbb{E}_{y \sim \pi(\cdot \mid x_0)}[\nabla \log \pi(y \mid x_0)] = \nabla \int \pi(y \mid x_0) \, dy = 0,$$

where the first equality is the standard importance-sampling identity under $\pi_{\text{ref}}(\cdot) = \pi(\cdot \mid x_1^i)$. Hence every cross term vanishes. We conclude

$$\mathbb{E}\left[\frac{\pi(y_1^i \mid x_0)}{\pi(y_1^i \mid x_1^i)} A_i^{(1)} \nabla \log \pi(y_1^i \mid x_0) \mid x_0\right] = \left[1 - \frac{1}{N}\right] \mathbb{E}\left[\frac{\pi(y_1^i \mid x_0)}{\pi(y_1^i \mid x_1^i)} r_1^i \nabla \log \pi(y_1^i \mid x_0) \mid x_0\right].$$

Finally, by the unbiasedness of the importance-corrected estimator with $A = r_1$,

$$\mathbb{E}\left[\frac{\pi(y_1^i \mid x_0)}{\pi(y_1^i \mid x_1^i)} r_1^i \nabla \log \pi(y_1^i \mid x_0) \mid x_0\right] = \nabla \mathbb{E}_{y \sim \pi(\cdot \mid x_0)}[r(x_0, y)].$$

Averaging over $i$ gives the claimed result. $\qquad\square$

**Remark.** Replacing $\bar{r}_1$ by the leave-one-out baseline $\bar{r}_{1,-i} = \frac{1}{N-1} \sum_{j \neq i} r_1^j$ eliminates the self-coupling term and yields $\mathbb{E}[\hat{G}^{(2)} \mid x_0] = \nabla \mathbb{E}_{y \sim \pi(\cdot \mid x_0)}[r(x_0, y)]$.

**Proposition C.2** (First-turn group-mean baseline is unbiased (with IS correction)). *For a fixed $x_0$, define the first-turn mean baseline*

$$\bar{r}_0 := \frac{1}{N} \sum_{j=1}^{N} r_0^j, \qquad A_i^{(0)} := r_1^i - \bar{r}_0.$$

*Consider the importance-corrected gradient estimator*

$$\hat{G}^{(1)} := \frac{1}{N} \sum_{i=1}^{N} \frac{\pi(y_1^i \mid x_0)}{\pi(y_1^i \mid x_1^i)} A_i^{(0)} \nabla \log \pi(y_1^i \mid x_0).$$

*Then, conditioning on $x_0$,*

$$\mathbb{E}\left[\hat{G}^{(1)} \mid x_0\right] = \nabla \mathbb{E}_{y \sim \pi(\cdot \mid x_0)}[r(x_0, y)].$$

*In other words, the first-turn mean baseline does not introduce bias in expectation.*

**Proof.** Fix $x_0$ and an index $i$. Expanding the expectation,

$$\mathbb{E}\left[\frac{\pi(y_1^i \mid x_0)}{\pi(y_1^i \mid x_1^i)} (r_1^i - \bar{r}_0) \nabla \log \pi(y_1^i \mid x_0) \mid x_0\right]$$
$$= \mathbb{E}\left[\frac{\pi(y_1^i \mid x_0)}{\pi(y_1^i \mid x_1^i)} r_1^i \nabla \log \pi(y_1^i \mid x_0) \mid x_0\right] - \mathbb{E}\left[\bar{r}_0 \cdot \frac{\pi(y_1^i \mid x_0)}{\pi(y_1^i \mid x_1^i)} \nabla \log \pi(y_1^i \mid x_0) \mid x_0\right].$$

The first term equals the desired gradient by importance-sampling correction:

$$\mathbb{E}\left[\frac{\pi(y_1^i \mid x_0)}{\pi(y_1^i \mid x_1^i)} r_1^i \nabla \log \pi(y_1^i \mid x_0) \mid x_0\right] = \nabla \mathbb{E}_{y \sim \pi(\cdot \mid x_0)}[r(x_0, y)].$$

For the second term, condition on the $\sigma$-field generated by the first-turn variables $\mathcal{H} := \sigma(\{y_0^j, c_0^j, x_1^j\}_{j=1}^{N})$. Then $\bar{r}_0$ is $\mathcal{H}$-measurable, and given $\mathcal{H}$ the only randomness in the $i$-th factor is $y_1^i \sim \pi(\cdot \mid x_1^i)$. Hence,

$$\mathbb{E}\left[\bar{r}_0 \cdot \frac{\pi(y_1^i \mid x_0)}{\pi(y_1^i \mid x_1^i)} \nabla \log \pi(y_1^i \mid x_0) \mid x_0\right] = \mathbb{E}\left[\bar{r}_0 \cdot \mathbb{E}\left[\frac{\pi(y_1^i \mid x_0)}{\pi(y_1^i \mid x_1^i)} \nabla \log \pi(y_1^i \mid x_0) \mid x_0, \mathcal{H}\right] \mid x_0\right].$$

By importance sampling,

$$\mathbb{E}\left[\frac{\pi(y_1^i \mid x_0)}{\pi(y_1^i \mid x_1^i)} \nabla \log \pi(y_1^i \mid x_0) \mid x_0, \mathcal{H}\right] = \mathbb{E}_{y \sim \pi(\cdot \mid x_0)}[\nabla \log \pi(y \mid x_0)] = \nabla \int \pi(y \mid x_0)\, dy = 0.$$

Therefore the entire second term is zero, and we obtain

$$\mathbb{E}\left[\frac{\pi(y_1^i \mid x_0)}{\pi(y_1^i \mid x_1^i)} (r_1^i - \bar{r}_0) \nabla \log \pi(y_1^i \mid x_0) \mid x_0\right] = \nabla \mathbb{E}_{y \sim \pi(\cdot \mid x_0)}[r(x_0, y)].$$

Averaging over $i$ yields the claim. $\qquad\square$

## C.2. Difference between unbiasedness and point-wise signal collapse

Fix a prompt $x_0$ and a group of $N$ second-round samples $\{y_1^i\}_{i=1}^N$ with rewards $r_1^i := r(x_0, y_1^i) \in [0, 1]$. Consider a generic (possibly importance-corrected) estimator

$$\hat{g} = \frac{1}{N} \sum_{i=1}^N w_i A_i s_i, \qquad s_i := \nabla \log \pi(y_1^i \mid x_0),$$

where $w_i$ is a weight (e.g. $w_i = \pi(y_1^i \mid x_0)/\pi(y_1^i \mid x_1^i)$) and $A_i$ is an advantage.

**Unbiasedness is an in-expectation statement.** Under the assumptions in Appendix B, for suitable choices of $A_i$ and $w_i$ we have $\mathbb{E}[\hat{g} \mid x_0] = \nabla J(\pi)$ (e.g. Eq. (10)), which is a statement about the conditional mean.

**Signal collapse is a point-wise (distributional) statement.** Even if $\mathbb{E}[\hat{g} \mid x_0]$ matches the target gradient, $\hat{g}$ can be identically zero for a nontrivial fraction of groups, yielding no update on those groups.

**Proposition C.3** (Deterministic collapse under second-turn mean baselines). *Let the second-turn baseline be a mean of the same group rewards, e.g. the in-sample mean $\bar{r}_1 := \frac{1}{N} \sum_{j=1}^N r_1^j$ or the leave-one-out mean $\bar{r}_{1,-i} := \frac{1}{N-1} \sum_{j \neq i} r_1^j$, and define $A_i^{(1)} := r_1^i - \bar{r}_1$ (or $r_1^i - \bar{r}_{1,-i}$). If the group rewards are constant, i.e. $r_1^1 = \cdots = r_1^N$, then $A_i^{(1)} \equiv 0$ for all $i$ and hence $\hat{g} \equiv 0$, regardless of the weights $\{w_i\}$.*

**Proof.** If $r_1^1 = \cdots = r_1^N$, then $\bar{r}_1 = r_1^i$ and also $\bar{r}_{1,-i} = r_1^i$ for each $i$. Thus each advantage equals zero deterministically, so every summand vanishes. $\qquad \square$

**Proposition C.4** (How often collapse occurs for Bernoulli rewards). *Assume $r_1^i \in \{0, 1\}$ are i.i.d. given $x_0$ with $\Pr(r_1^i = 1 \mid x_0) = p_1$. Under either in-sample or leave-one-out second-turn mean baselines, the estimator collapses with probability*

$$\Pr(\hat{g} = 0 \mid x_0) \geq \Pr(r_1^1 = \cdots = r_1^N \mid x_0) = p_1^N + (1 - p_1)^N.$$

*In particular, when $p_1 \to 1$, the probability of a* nonzero *update scales as $1 - p_1^N \approx N(1 - p_1)$.*

**Near-collapse and variance interpretation.** Let $\bar{r}_1 = \frac{1}{N} \sum_i r_1^i$. For the in-sample mean baseline $A_i^{(1)} = r_1^i - \bar{r}_1$,

$$\frac{1}{N} \sum_{i=1}^N (A_i^{(1)})^2 = \frac{1}{N} \sum_{i=1}^N (r_1^i - \bar{r}_1)^2$$

is exactly the empirical variance of the group rewards. Thus whenever the second-turn rewards concentrate (e.g. high success-rate distillation where $r_1^i \approx 1$), the advantages are uniformly small and the update magnitude is small. This complements the in-expectation analysis in Appendix B: unbiasedness controls the mean of $\hat{g}$, whereas collapse is governed by the *mass of $\hat{g}$ near zero*, which can be large when rewards saturate.

## C.3. Discussion on Alternative Baselines

A natural alternative to the first-turn mean baseline is the *trajectory-level improvement* advantage

$$A_i^\Delta := R(x_0, y_1^i) - R(x_0, y_0^i), \tag{9}$$

which measures how much the critique-conditioned revision improves a particular trajectory. This choice is unbiased in our estimator because $R(x_0, y_0^i)$ does not depend on the scored action $y_1^i$.

**Variance comparison.** To compare variance, it is helpful to separate the effect of the advantage from the score term. Fix a prompt $x_0$ and consider the conditional variance of the scalar advantage (the same comparison carries through to the gradient in any fixed direction if the score is uniformly bounded). Let

$$R_1 := R(x_0, y_1), \qquad R_0 := R(x_0, y_0),$$

and let $b^{(0)} = \frac{1}{N} \sum_{j=1}^N R(x_0, y_0^j)$ be the first-turn mean baseline used in (6). For a single trajectory $i$, define

$$A^{(0)} := R_1 - b^{(0)}, \qquad A^\Delta := R_1 - R_0.$$

Conditioned on $x_0$, $R_1$ is independent of the *other* first-turn samples $\{R(x_0, y_0^j)\}_{j \neq i}$, hence independent of $b^{(0)}$ up to a $1/N$ self-term. Approximating this finite-sample effect by ignoring the self-term (or using the leave-one-out baseline), we have $\text{Cov}(R_1, b^{(0)} \mid x_0) \approx 0$ and therefore

$$\text{Var}(A^{(0)} \mid x_0) = \text{Var}(R_1 \mid x_0) + \text{Var}(b^{(0)} \mid x_0) \approx \text{Var}(R_1 \mid x_0) + \frac{1}{N}\text{Var}(R_0 \mid x_0), \tag{10}$$

$$\text{Var}(A^\Delta \mid x_0) = \text{Var}(R_1 - R_0 \mid x_0) = \text{Var}(R_1 \mid x_0) + \text{Var}(R_0 \mid x_0) - 2\text{Cov}(R_1, R_0 \mid x_0). \tag{11}$$

Subtracting (10) from (11) yields

$$\text{Var}(A^\Delta \mid x_0) - \text{Var}(A^{(0)} \mid x_0) \approx \left(1 - \frac{1}{N}\right)\text{Var}(R_0 \mid x_0) - 2\text{Cov}(R_1, R_0 \mid x_0). \tag{12}$$

Equation (12) shows that $A^\Delta$ has *larger* conditional variance than $A^{(0)}$ whenever

$$\text{Cov}(R_1, R_0 \mid x_0) \leq \frac{1}{2}\left(1 - \frac{1}{N}\right)\text{Var}(R_0 \mid x_0). \tag{13}$$

This is the typical regime in our setting. In sparse-reward problems, $R_0$ is near-deterministically zero under the base policy (so $\text{Var}(R_0 \mid x_0) \approx p_0(x_0)$), while the dependence between first-turn success and post-feedback success is often weak or even *negative*: critiques primarily help when the first attempt fails, making $R_1$ and $R_0$ less positively correlated. In particular, if $\text{Cov}(R_1, R_0 \mid x_0) \approx 0$ (or $\leq 0$), then

$$\text{Var}(A^\Delta \mid x_0) \gtrsim \text{Var}(A^{(0)} \mid x_0) + \left(1 - \frac{1}{N}\right)\text{Var}(R_0 \mid x_0),$$

so the improvement baseline pays an extra variance term of order $\text{Var}(R_0)$, whereas the mean baseline only pays $\text{Var}(R_0)/N$.

**Additional downsides beyond variance.** The improvement baseline also changes *what* the algorithm emphasizes:

- **It discards many informative second-turn successes.** If both attempts succeed ($R_0 = R_1 = 1$), then $A^\Delta = 0$ and the trajectory contributes no learning signal, even though $y_1$ may still contain useful "clean" solutions worth distilling into the one-shot policy. In contrast, $A^{(0)} = 1 - b^{(0)}$ remains positive whenever the first-turn policy is imperfect ($b^{(0)} < 1$), so it continues to reinforce successful corrected outputs.
- **It provides weak normalization across prompts.** When $R_0 \approx 0$ for most samples, $A^\Delta \approx R_1$ and the method effectively reduces to using raw post-feedback rewards, losing the prompt-level normalization that makes (6) stable when post-feedback success becomes high.

## C.4. Recovering Rejection Sampling

Our framework recovers the commonly used Rejection Sampling (or SFT) baseline in distillation: namely, we collect correct second-round generations and then SFT them into the single-turn policy $\pi(\cdot \mid x_0)$. Concretely, for each $x_0$ we sample $y_0 \sim \pi(\cdot \mid x_0)$, form $x_1 = f(x_0, y_0, c_0)$, and sample $y_1 \sim \pi(\cdot \mid x_1)$. We keep $y_1$ if it is correct, and perform maximum likelihood on the accepted set.

This procedure fits into Eq. (3) by taking a binary advantage $A(y_1) = R(x_0, y_1) \in \{0, 1\}$, and choosing the reference as the same single-turn policy, but treated as a constant via stop-gradient, i.e., $\pi_{\text{ref}}(\cdot \mid x_0) = \text{sg}[\pi(\cdot \mid x_0)]$. With this choice, the ratio $\pi(y_1 \mid x_0)/\pi_{\text{ref}}(y_1 \mid x_0)$ evaluates to 1 in the forward pass while still producing the desired score-function gradient. In particular, the induced update direction becomes

$$\nabla \ell_{\text{distill}}^{\text{SFT}}(\pi) := \mathbb{E}_{y_1 \sim \pi(\cdot \mid x_1)}\left[\frac{\pi(y_1 \mid x_0)}{\text{sg}(\pi(y_1 \mid x_0))} R(x_0, y_1) \nabla \log \pi(y_1 \mid x_0)\right] = \mathbb{E}_{y_1 \sim \pi(\cdot \mid x_1)}[R(x_0, y_1) \nabla \log \pi(y_1 \mid x_0)].$$

Therefore, in our setting SFT is precisely the special case that *distills only the correct second-round generations* (via rejection sampling on the 0–1 reward), without any importance-weight variance from off-policy correction.

# D. Theory Results from Section 4

### D.1. Setup

We analyze an early-stage (batch) regime of RL post-training through a horizon-1 contextual bandit abstraction. A prompt (context) is sampled as $x \sim \mu$, the model outputs a response $y \in \mathcal{Y}(x)$, and receives a bounded reward $R(x, y) \in [0, 1]$. The objective is

$$J(\pi) := \mathbb{E}_{x \sim \mu} \mathbb{E}_{y \sim \pi(\cdot \mid x)}[R(x, y)].$$

**Log-linear policy with learned representation.** We parameterize the policy by a learned representation $z_w(x, y) \in \mathbb{R}^d$ and a linear head $b \in \mathbb{R}^d$, with parameters $\theta = (b, w)$. Define the score function

$$f_\theta(x, y) := b^\top z_w(x, y),$$

and the log-linear policy

$$\pi_\theta(y \mid x) := \frac{\exp(\tau f_\theta(x, y))}{\sum_{y' \in \mathcal{Y}(x)} \exp(\tau f_\theta(x, y'))}, \tag{14}$$

where $\tau > 0$ is an inverse temperature.

**Frozen rollout distribution.** We study an early-stage regime in which rollouts remain close to the base policy, so samples are effectively drawn from a fixed distribution

$$d_0(x, y) := \mu(x) \, \pi_{\theta_0}(y \mid x), \qquad \theta_0 = (b_0, w_0).$$

Equivalently, we take $\pi_{\text{base}} := \pi_{\theta_0}$. This frozen-rollout assumption is the standing regime for the directional SNR calculations in Appendix D.1 and the trajectory bounds in Appendix D.2.

**REINFORCE estimator and score features.** Let

$$s_\theta(x, y) := \nabla_\theta \log \pi_\theta(y \mid x), \qquad g(x, y) := s_{\theta_0}(x, y)$$

denote the score at initialization. The reward-only REINFORCE estimator at $\theta_0$ is

$$\widehat{g}(x, y) := R(x, y) \, g(x, y), \qquad (x, y) \sim d_0,$$

and the corresponding population gradient is

$$g_{\text{RL}}(\theta_0) := \nabla_\theta J(\pi_\theta)\big|_{\theta=\theta_0} = \mathbb{E}_{(x,y) \sim d_0}[R(x, y) \, g(x, y)].$$

**Fisher information and reward-weighted second moment.** Define the (rollout) Fisher information matrix at $\theta_0$ by

$$I(\theta_0) := \mathbb{E}_{d_0}[g(x, y)g(x, y)^\top].$$

For the linear head $b$, letting

$$\phi_{\theta_0}(x, y) := z_{w_0}(x, y) - \mathbb{E}_{y' \sim \pi_{\theta_0}(\cdot \mid x)}[z_{w_0}(x, y')],$$

we have the closed-form score

$$\nabla_b \log \pi_{\theta_0}(y \mid x) = \tau \, \phi_{\theta_0}(x, y),$$

so the Fisher restricted to $b$ equals the policy-induced feature covariance:

$$I_b(\theta_0) = \tau^2 \, \mathbb{E}_{x \sim \mu}\left[\text{Cov}_{y \sim \pi_{\theta_0}(\cdot \mid x)}\left(z_{w_0}(x, y)\right)\right].$$

We will also use the reward-weighted score second moment

$$\Sigma_{\text{RL}}(\theta_0) := \mathbb{E}_{(x,y) \sim d_0}[R(x, y)^2 \, g(x, y)g(x, y)^\top]. \tag{15}$$

For any unit direction $u$, the second moment of the directional estimator equals

$$M_{2,u} := \mathbb{E}[\langle \widehat{g}, u \rangle^2] = u^\top \Sigma_{\text{RL}}(\theta_0) \, u.$$

**Directional SNR.** Fix any unit direction $u$ and define

$$Z_u := \langle \widehat{g}, u \rangle = R(x,y) \langle g(x,y), u \rangle.$$

Let $\mu_u := \mathbb{E}[Z_u]$ and $M_{2,u} := \mathbb{E}[Z_u^2]$ under $(x,y) \sim d_0$. The per-sample directional signal-to-noise ratio is

$$\text{SNR}(u) := \frac{|\mu_u|}{\sqrt{M_{2,u}}}. \tag{16}$$

This quantity controls the sample complexity required to reliably estimate a gradient component along direction $u$.

**Lemma D.1** (Directional concentration in terms of SNR). *Let $Z_{u,1}, \ldots, Z_{u,N}$ be i.i.d. samples of $Z_u$ and $\overline{Z}_u := \frac{1}{N} \sum_{i=1}^{N} Z_{u,i}$. Assume $R \in [0,1]$ and $|\langle g(x,y), u \rangle| \le G_u$ almost surely under $d_0$, so $|Z_u| \le G_u$. Then for any $\delta \in (0,1)$, with probability at least $1 - \delta$,*

$$|\overline{Z}_u - \mu_u| \le \sqrt{\frac{2 \, M_{2,u} \, \log(2/\delta)}{N}} + \frac{4 G_u \log(2/\delta)}{3N}. \tag{17}$$

*Consequently, for any constant $\alpha \in (0,1)$, obtaining $|\overline{Z}_u - \mu_u| \le \alpha|\mu_u|$ (and hence recovering the sign of $\mu_u$ when $\alpha < 1$) requires*

$$N = \Omega\left( \frac{1}{\text{SNR}(u)^2} \log \frac{1}{\delta} \right),$$

*up to constant factors and lower-order $1/N$ terms.*

### D.2. Reward-only RL under base rollouts

We present two complementary bottlenecks for reward-only policy gradients under the frozen rollout distribution $d_0(x,y) = \mu(x)\pi_{\theta_0}(y \mid x)$. Both bottlenecks are stated in terms of the directional statistics in Appendix D.1: $Z_u = R\langle g, u \rangle$, $\mu_u = \mathbb{E}[Z_u]$, $M_{2,u} = \mathbb{E}[Z_u^2]$, and $\text{SNR}(u) = |\mu_u|/\sqrt{M_{2,u}}$.

D.2.1. RARE-EVENT REGIME: SMALL SUCCESS PROBABILITY IMPLIES LOW DIRECTIONAL SNR

We first formalize the common regime where reward is supported on a low-probability success event.

**Lemma D.2** (SNR bound under reward supported on a rare event). *Let $S(x,y) \in \{0,1\}$ be any event and define $\varepsilon_0 := \Pr_{(x,y) \sim d_0}(S = 1)$. Assume the reward is supported on S, i.e., $R(x,y) = 0$ whenever $S(x,y) = 0$. Then for any unit direction $u$,*

$$\text{SNR}(u) = \sqrt{\varepsilon_0} \cdot \frac{\left| \mathbb{E}[R(x,y)\langle g(x,y), u \rangle \mid S = 1] \right|}{\sqrt{\mathbb{E}[R(x,y)^2 \langle g(x,y), u \rangle^2 \mid S = 1]}} \le \sqrt{\varepsilon_0}.$$

**Corollary D.1** (Sample complexity under small pass rate). *Assume binary reward $R(x,y) = \mathbf{1}\{\text{pass}(x,y)\}$ and let $S(x,y) = \mathbf{1}\{\text{pass}(x,y)\}$. Then $\varepsilon_0 = \Pr_{(x,y) \sim d_0}(\text{pass} = 1)$ is the base pass rate and the assumption of Lemma D.2 holds. Hence for all unit $u$, $\text{SNR}(u) \le \sqrt{\varepsilon_0}$. Consequently, with probability at least $1 - \delta$, recovering $\text{sign}(\mu_u)$ for any unit direction $u$ requires*

$$N = \Omega\left( \frac{1}{\varepsilon_0} \log \frac{1}{\delta} \right)$$

rollouts.

D.2.2. WEAK IDENTIFIABILITY OF REPRESENTATION DIRECTIONS UNDER SUCCESS CONDITIONING

**Motivation.** Corollary D.1 is a finite-sample statement: when successes are rare, large batches are required to reliably estimate gradient components. We now isolate a population-level geometric limitation that can persist even with access to the exact population gradient under $d_0$, focusing on the representation parameters $w$ with $b = b_0$ fixed.

**Assumption D.1** (Frozen rollout distribution for the first $T$ steps). *For the first $T$ gradient steps, all expectations defining the update are taken under the same fixed distribution $(x,y) \sim d_0(x,y) = \mu(x)\pi_{\theta_0}(y \mid x)$, and $R(x,y) = \mathbf{1}\{\text{pass}(x,y)\}$ is evaluated on samples from $d_0$.*

Define the representation score (with head fixed at $b = b_0$)

$$s_w^w(x,y) := \nabla_w \log \pi_{(b_0,w)}(y \mid x) \in \mathbb{R}^p, \qquad \varepsilon_0 := \Pr_{(x,y)\sim d_0}(\text{pass}(x,y) = 1),$$

and the success-conditioned representation score second moment under $d_0$

$$\Sigma_{\text{succ}}^w(w) := \mathbb{E}\big[s_w^w(x,y)\, s_w^w(x,y)^\top \,\big|\, \text{pass}(x,y) = 1\big] \in \mathbb{R}^{p\times p}.$$

Let $\Sigma_{\text{succ}}^w(w_0) = \sum_{i=1}^m \lambda_i v_i v_i^\top$ with $\lambda_1 \geq \cdots \geq \lambda_m \geq 0$ and define the low-signal subspace and projector

$$S_{\text{low}}(\eta) := \text{span}\{v_i : \lambda_i < \eta\}, \qquad \Pi := \Pi_{S_{\text{low}}(\eta)}.$$

**Interpretation.** The spectrum of $\Sigma_{\text{succ}}^w(w_0)$ quantifies which representation directions are statistically identifiable from success-conditioned policy scores under base rollouts. Small eigenvalues indicate directions in which even successful samples carry little score second moment, so reward-weighted updates have negligible projection onto those directions at initialization under $d_0$.

**Lemma D.3** (Projected gradient bound under fixed $d_0$). *Adopt Assumption D.1 and consider the fixed-$d_0$ reward gradient field*

$$g_{\text{RL}}^w(w) := \mathbb{E}_{(x,y)\sim d_0}[R(x,y)\, s_w^w(x,y)].$$

*Then for any $w$,*

$$\|\Pi g_{\text{RL}}^w(w)\|_2 \ \leq\ \varepsilon_0 \sqrt{\ \|\Pi\, \Sigma_{succ}^w(w)\, \Pi\|_{\text{op}}\ }.$$

*In particular, $\|\Pi g_{\text{RL}}^w(w_0)\|_2 \leq \varepsilon_0 \sqrt{\eta}$.*

**Theorem D.1** (Low-signal progress bound under fixed $d_0$). *Adopt Assumption D.1 and consider exact gradient ascent updates under fixed $d_0$:*

$$w_{t+1} = w_t + \rho\, g_{\text{RL}}^w(w_t), \qquad g_{\text{RL}}^w(w) := \mathbb{E}_{(x,y)\sim d_0}[R(x,y)\, s_w^w(x,y)].$$

*Then for any integer $T \geq 1$,*

$$\|\Pi(w_T - w_0)\|_2 \ \leq\ \rho\, \varepsilon_0 \sum_{t=0}^{T-1} \sqrt{\ \|\Pi\, \Sigma_{succ}^w(w_t)\, \Pi\|_{\text{op}}\ }.$$

**Remark D.1.** Theorem D.1 is a statement about the representation parameters $w$ with the head held fixed at $b = b_0$. It does not rule out the possibility that optimization may also be limited by the linear head $b$ (or by joint $(b,w)$ interactions) in other regimes. Our purpose here is narrower: we isolate representation directions that are weakly identified by reward-only learning under base rollouts, to compare with the representation movement induced by RLTF-FM in Appendix D.3.

**Remark D.2** (Interpretation). Under frozen base rollouts $(x,y) \sim d_0$, projected progress in the low-signal subspace $S_{\text{low}}(\eta)$ is controlled by the cumulative success-conditioned score second moment along the trajectory,

$$\sum_{i<t} \sqrt{\|\Pi\Sigma_{\text{succ}}^w(w_i)\Pi\|_{\text{op}}}.$$

Thus, even with a fixed data distribution, an early-stage plateau can occur when successful samples have small success-conditioned score second moment in $S_{\text{low}}(\eta)$ over the window of interest.

**Corollary D.2** (An illustrative plateau condition). Adopt the assumptions of Theorem D.1. Fix any $\Delta > 0$ and define the superlevel set

$$\Theta_\Delta := \big\{w : J(\pi_{(b_0,w)}) \geq J(\pi_{(b_0,w_0)}) + \Delta\big\}.$$

Define the required displacement in the low-signal subspace by

$$r_\Delta := \inf_{w\in\Theta_\Delta} \|\Pi(w - w_0)\|_2.$$

Assume $r_\Delta > 0$ and define the trajectory-dependent cumulative score second moment quantity

$$\mathcal{E}_t := \sum_{i=0}^{t-1} \sqrt{\ \|\Pi\, \Sigma_{\text{succ}}^w(w_i)\, \Pi\|_{\text{op}}\ }.$$

Then for any $t \leq T$, if $\rho \, \varepsilon_0 \, \mathcal{E}_t < r_\Delta$, the iterates satisfy

$$J(\pi_{(b_0, w_t)}) \; < \; J(\pi_{(b_0, w_0)}) + \Delta.$$

Equivalently, achieving $J(\pi_{(b_0, w_t)}) \geq J(\pi_{(b_0, w_0)}) + \Delta$ within this regime requires $\rho \, \varepsilon_0 \, \mathcal{E}_t \geq r_\Delta$.

**Remark D.3.** The condition $r_\Delta > 0$ is problem-dependent: it asserts that achieving a $\Delta$ improvement in expected reward with fixed head $b_0$ requires nontrivial movement in the low-signal subspace as measured by $\|\Pi(w - w_0)\|_2$. We include Corollary D.2 to make explicit how a low-signal representation subspace can induce an early-stage plateau under base rollouts, in terms of the cumulative success-conditioned score second moment $\mathcal{E}_t$ supplied by successful samples.

### D.3. Feedback modeling: representation learning benefit in the RL low-signal subspace

We isolate a representation-learning benefit of RLTF-FM in the same early-stage frozen-rollout regime as Appendix D.2. We do not prove reward improvement here. Instead, we formalize the following claim:

*Under base rollouts, reward-only learning can have negligible driving signal in a low-signal representation subspace $S_{\text{low}}(\eta)$ defined in Appendix D.2. In contrast, RLTF-FM provides auxiliary supervision that induces nontrivial representation movement in this subspace, so these degrees of freedom become statistically identifiable even before the rollout distribution shifts.*

#### D.3.1. SETUP

We model a shared representation $z_w(x, y) \in \mathbb{R}^d$ with $w \in \mathbb{R}^p$. Write its Jacobian at initialization as

$$J(x, y) \; := \; \nabla_w z_w(x, y)\big|_{w=w_0} \in \mathbb{R}^{d \times p},$$

so that $J(x, y)^\top v \in \mathbb{R}^p$ for any $v \in \mathbb{R}^d$.

**Policy score in representation space.** Holding the head fixed at $b = b_0$, the policy representation score for Eq. (14) is

$$s_w^w(x, y) = \nabla_w \log \pi_{(b_0, w)}(y \mid x) = \tau \Big( J(x, y)^\top b_0 - \mathbb{E}_{y' \sim \pi_{(b_0, w)}(\cdot \mid x)}[J(x, y')^\top b_0] \Big).$$

In particular, at $w_0$ this matches the score used to define $\Sigma_{\text{succ}}^w(w_0)$ and $S_{\text{low}}(\eta)$ in Appendix D.2.

**Feedback model.** Let $\mathcal{C}$ be the set of critique token/types and let $u_\psi(c) \in \mathbb{R}^d$ be a class embedding for critique type $c$, parameterized by $\psi$. Define the feedback model

$$p_{\psi, w}(c \mid x, y) \; := \; \frac{\exp\big(\langle u_\psi(c), z_w(x, y)\rangle\big)}{\sum_{c' \in \mathcal{C}} \exp\big(\langle u_\psi(c'), z_w(x, y)\rangle\big)}.$$

Its representation score at $(\psi_0, w_0)$ is

$$s_{\text{FM}}(x, y, c) := \nabla_w \log p_{\psi_0, w}(c \mid x, y)\big|_{w=w_0} = J(x, y)^\top \Big( u_{\psi_0}(c) - \mathbb{E}_{c' \sim p_{\psi_0, w_0}(\cdot \mid x, y)}[u_{\psi_0}(c')] \Big) \in \mathbb{R}^p.$$

All expectations below are under $(x, y) \sim d_0 := \mu(x)\pi_{\theta_0}(y \mid x)$ and $c \sim \mathcal{M}(\cdot \mid x, y)$.

Recall from Appendix D.2 that $S_{\text{low}}(\eta) \subseteq \mathbb{R}^p$ and $\Pi \in \mathbb{R}^{p \times p}$ are defined via the success-conditioned second moment $\Sigma_{\text{succ}}^w(w_0)$ of the policy representation score.

#### D.3.2. ASSUMPTIONS

Define the FM score mean and centered covariance at $(\psi_0, w_0)$:

$$m_{\text{FM}} := \mathbb{E}[s_{\text{FM}}(x, y, c)], \qquad C_{\text{FM}} := \mathbb{E}\big[(s_{\text{FM}}(x, y, c) - m_{\text{FM}})(s_{\text{FM}}(x, y, c) - m_{\text{FM}})^\top\big].$$

**Assumption D.2** (FM drift in the RL low-signal subspace)**.** *There exists $b_{\text{FM}} > 0$ such that $\|\Pi m_{\text{FM}}\|_2 \geq b_{\text{FM}}$.*

A sufficient interpretation is that, under base rollouts, the feedback model $p_{\psi_0, w_0}$ is moment-mismatched with the feeder $\mathcal{M}$ along directions in $S_{\text{low}}(\eta)$. Indeed,

$$m_{\text{FM}} = \mathbb{E}_{(x,y) \sim d_0}\Big[ J(x,y)^\top \Big( \mathbb{E}_{c \sim \mathcal{M}(\cdot|x,y)}[u_{\psi_0}(c)] - \mathbb{E}_{c \sim p_{\psi_0, w_0}(\cdot|x,y)}[u_{\psi_0}(c)] \Big) \Big].$$

**Assumption D.3** (FM coverage). *There exists $\gamma_{\text{FM}} > 0$ such that*

$$\Pi \, C_{\text{FM}} \, \Pi \succeq \gamma_{\text{FM}} \Pi.$$

This assumption is a covariance conditioning requirement: the feedback score covariance is non-degenerate on $S_{\text{low}}(\eta)$ under base rollouts, so directions in this subspace are statistically identifiable from feedback supervision.

### D.3.3. RESULT: REPRESENTATION-LEARNING BENEFIT

We analyze FM-only updates on the shared representation parameters using the frozen score evaluated at $w_0$:

$$w_{t+1} = w_t + \rho\lambda \, s_{\text{FM}}(x_t, y_t, c_t)\big|_{w=w_0}, \qquad (x_t, y_t) \sim d_0, \ \ c_t \sim \mathcal{M}(\cdot \mid x_t, y_t),$$

with step size $\rho > 0$ and FM weight $\lambda > 0$.

**Theorem D.2** (FM moves the shared representation in the RL low-signal subspace). *Let $k := \text{tr}(\Pi) = \dim(S_{\text{low}}(\eta))$. For any integer $T \geq 1$,*

$$\mathbb{E}\big[\|\Pi(w_T - w_0)\|_2^2\big] = \rho^2\lambda^2\Big(T \, \text{tr}(\Pi C_{\text{FM}}\Pi) \ + \ T^2 \, \|\Pi m_{\text{FM}}\|_2^2\Big).$$

*In particular, under Assumptions D.2 and D.3,*

$$\mathbb{E}\big[\|\Pi(w_T - w_0)\|_2^2\big] \ \geq \ \rho^2\lambda^2\Big(T \, \gamma_{\text{FM}} \, k \ + \ T^2 \, b_{\text{FM}}^2\Big).$$

**Remark D.4.** Theorem D.2 decomposes movement in $S_{\text{low}}(\eta)$ into a covariance term $T \, \text{tr}(\Pi C_{\text{FM}}\Pi)$ (coverage) and a mean-drift term $T^2\|\Pi m_{\text{FM}}\|_2^2$ (moment mismatch). For large $T$, the drift term dominates whenever $\|\Pi m_{\text{FM}}\|_2$ is not extremely small, so systematic feeder/model moment mismatch can be the primary driver of representation movement in this regime. Coverage becomes most important when drift is weak (e.g., after partial fitting of the feedback model) or when one seeks broadly conditioned updates across $S_{\text{low}}(\eta)$ rather than movement concentrated along a single biased direction.

**Remark D.5.** Theorem D.2 is a frozen-score (linearized) calculation: it characterizes the initial FM signal under samples from $d_0$ by evaluating the score at $w_0$. In contrast, the reward-only analysis in Theorem D.1 is stated along the optimization trajectory $w_t$ (under the same frozen rollout distribution $d_0$), and it upper bounds projected progress by a path integral of success-conditioned score second moment. Thus, these results serve different roles rather than forming symmetric global convergence statements: Theorem D.2 establishes that the FM vector field has nontrivial components in $S_{\text{low}}(\eta)$ at initialization (signal availability), whereas Theorem D.1 shows that reward-only progress in that subspace is limited unless successful samples carry substantial success-conditioned score second moment in those directions along the trajectory. Extending Theorem D.2 to a trajectory-level FM analysis would require controlling how the FM score statistics evolve as $w$ changes.

**Remark D.6** (Representation-learning benefit under base rollouts). Appendix D.2 identifies $S_{\text{low}}(\eta)$ (defined from $\Sigma_{\text{succ}}^w(w_0)$) as representation directions that can be weakly identified by reward-only learning under base rollouts. Theorem D.2 shows that, in the same frozen distribution regime, RLTF-FM induces nontrivial representation movement along these directions through a systematic mean component $\Pi m_{\text{FM}}$ and covariance conditioning $\Pi C_{\text{FM}}\Pi$. In this sense, RLTF-FM can make low-signal representation degrees of freedom statistically identifiable from feedback supervision before any reward-improvement guarantee is invoked.

### D.4. Proofs

**Proof of Lemma D.1.** Let $X_i := Z_{u,i} - \mu_u$. Then $\mathbb{E}[X_i] = 0$ and

$$|X_i| \leq |Z_{u,i}| + |\mu_u| \leq G_u + \mathbb{E}|Z_u| \leq 2G_u \qquad \text{a.s.}$$

Moreover, $\mathrm{Var}(X_i) = \mathrm{Var}(Z_u) \leq \mathbb{E}[Z_u^2] = M_{2,u}$. By Bernstein's inequality for bounded i.i.d. mean-zero variables, for any $\delta \in (0,1)$, with probability at least $1 - \delta$,

$$\left| \frac{1}{N} \sum_{i=1}^{N} X_i \right| \leq \sqrt{\frac{2\,\mathrm{Var}(Z_u)\,\log(2/\delta)}{N}} + \frac{2(2G_u)\log(2/\delta)}{3N}.$$

Since $\frac{1}{N}\sum_{i=1}^{N} X_i = \overline{Z}_u - \mu_u$, and $\mathrm{Var}(Z_u) \leq M_{2,u}$, this yields (17).

For the relative-error condition, it suffices that each term on the right-hand side of (17) is at most $\frac{\alpha}{2}|\mu_u|$. The square-root term condition gives

$$\sqrt{\frac{2M_{2,u}\log(2/\delta)}{N}} \leq \frac{\alpha}{2}|\mu_u| \quad \Longleftrightarrow \quad N \geq \frac{8M_{2,u}}{\alpha^2 \mu_u^2}\log\frac{2}{\delta}.$$

The linear term condition gives

$$\frac{4G_u\log(2/\delta)}{3N} \leq \frac{\alpha}{2}|\mu_u| \quad \Longleftrightarrow \quad N \geq \frac{8G_u}{3\alpha|\mu_u|}\log\frac{2}{\delta}.$$

Combining the two yields and substituting $\mathrm{SNR}(u) = |\mu_u|/\sqrt{M_{2,u}}$ gives the stated $\Omega(1/\mathrm{SNR}(u)^2)$ dependence. $\qquad\square$

**Proof of Lemma D.2.** Since $R(x,y) = 0$ on $\{S = 0\}$, we have $Z_u = 0$ on $\{S = 0\}$ as well, and hence

$$\mu_u = \mathbb{E}[Z_u] = \mathbb{E}[Z_u \mathbf{1}\{S = 1\}] = \mathrm{Pr}(S = 1) \cdot \mathbb{E}[Z_u \mid S = 1] = \varepsilon_0 \, \mathbb{E}[Z_u \mid S = 1].$$

Similarly,

$$M_{2,u} = \mathbb{E}[Z_u^2] = \mathbb{E}[Z_u^2 \mathbf{1}\{S = 1\}] = \varepsilon_0 \, \mathbb{E}[Z_u^2 \mid S = 1].$$

Therefore,

$$\mathrm{SNR}(u) = \frac{|\mu_u|}{\sqrt{M_{2,u}}} = \sqrt{\varepsilon_0} \cdot \frac{|\mathbb{E}[Z_u \mid S = 1]|}{\sqrt{\mathbb{E}[Z_u^2 \mid S = 1]}}.$$

By Jensen, $|\mathbb{E}[Z_u \mid S = 1]| \leq \sqrt{\mathbb{E}[Z_u^2 \mid S = 1]}$, so the ratio is at most 1. Substituting $Z_u = R(x,y)\langle g(x,y), u\rangle$ yields the displayed expression and the bound $\mathrm{SNR}(u) \leq \sqrt{\varepsilon_0}$. $\qquad\square$

**Proof of Corollary D.1.** The first claim follows immediately from Lemma D.2 by taking $S = \mathbf{1}\{\text{pass}\}$ and noting that binary $R$ is supported on $S$.

For the sample complexity statement, apply Lemma D.1 to the same scalar $Z_u$ with any constant $\alpha \in (0,1)$ (e.g., $\alpha = \frac{1}{2}$). Lemma D.1 states that achieving $|\overline{Z}_u - \mu_u| \leq \alpha|\mu_u|$ with probability at least $1 - \delta$ requires

$$N = \Omega\left( \frac{1}{\mathrm{SNR}(u)^2}\log\frac{1}{\delta} \right) \quad \text{(up to constant factors and lower-order terms).}$$

Since $\mathrm{SNR}(u) \leq \sqrt{\varepsilon_0}$, we have $1/\mathrm{SNR}(u)^2 \geq 1/\varepsilon_0$, yielding

$$N = \Omega\left( \frac{1}{\varepsilon_0}\log\frac{1}{\delta} \right).$$

Finally, when $\alpha < 1$, the condition $|\overline{Z}_u - \mu_u| \leq \alpha|\mu_u|$ implies $\overline{Z}_u$ has the same sign as $\mu_u$, so $\mathrm{sign}(\mu_u)$ is recovered. $\qquad\square$

**Proof of Lemma D.3.** Fix any unit $u \in \mathrm{range}(\Pi)$. Since $R = \mathbf{1}\{\text{pass}\}$,

$$\langle g_{\mathrm{RL}}^w(w), u\rangle = \mathbb{E}_{d_0}\big[R\,\langle s_w^w, u\rangle\big] = \varepsilon_0 \cdot \mathbb{E}\big[\langle s_w^w(x,y), u\rangle \mid \text{pass} = 1\big].$$

By Cauchy–Schwarz,

$$|\langle g_{\mathrm{RL}}^w(w), u\rangle| \leq \varepsilon_0\sqrt{\mathbb{E}\big[\langle s_w^w(x,y), u\rangle^2 \mid \text{pass} = 1\big]} = \varepsilon_0\sqrt{u^\top \Sigma_{\mathrm{succ}}^w(w)u}.$$

Taking the supremum over unit $u \in \mathrm{range}(\Pi)$ yields

$$\|\Pi g_{\mathrm{RL}}^w(w)\|_2 = \sup_{\substack{u \in \mathrm{range}(\Pi) \\ \|u\|_2 = 1}} \langle g_{\mathrm{RL}}^w(w), u \rangle \le \varepsilon_0 \sqrt{\|\Pi \Sigma_{\mathrm{succ}}^w(w) \Pi\|_{\mathrm{op}}} \,.$$

$\square$

**Proof of Theorem D.1.** By telescoping,

$$\Pi(w_T - w_0) = \rho \sum_{t=0}^{T-1} \Pi g_{\mathrm{RL}}^w(w_t),$$

so by triangle inequality,

$$\|\Pi(w_T - w_0)\|_2 \le \rho \sum_{t=0}^{T-1} \|\Pi g_{\mathrm{RL}}^w(w_t)\|_2.$$

Apply Lemma D.3 term-by-term concludes the proof. $\square$

**Proof of Corollary D.2.** By Theorem D.1, for any $t \le T$,

$$\|\Pi(w_t - w_0)\|_2 \ \le \ \rho \, \varepsilon_0 \sum_{i=0}^{t-1} \sqrt{\|\Pi \, \Sigma_{\mathrm{succ}}^w(w_i) \, \Pi\|_{\mathrm{op}}} \ = \ \rho \, \varepsilon_0 \, \mathcal{E}_t.$$

Thus, if $\rho \, \varepsilon_0 \, \mathcal{E}_t < r_\Delta$, then $\|\Pi(w_t - w_0)\|_2 < r_\Delta$.

By definition of $r_\Delta = \inf_{w \in \Theta_\Delta} \|\Pi(w - w_0)\|_2$, every $w \in \Theta_\Delta$ satisfies $\|\Pi(w - w_0)\|_2 \ge r_\Delta$. Therefore $\|\Pi(w_t - w_0)\|_2 < r_\Delta$ implies $w_t \notin \Theta_\Delta$, i.e.,

$$J(\pi_{(b_0, w_t)}) \ < \ J(\pi_{(b_0, w_0)}) + \Delta.$$

The final statement is just the contrapositive: if $J(\pi_{(b_0, w_t)}) \ge J(\pi_{(b_0, w_0)}) + \Delta$ then $w_t \in \Theta_\Delta$, so $\|\Pi(w_t - w_0)\|_2 \ge r_\Delta$, hence by the theorem $\rho \, \varepsilon_0 \, \mathcal{E}_t \ge r_\Delta$. $\square$

**Proof of Theorem D.2.** Let

$$s_t \ := \ s_{\mathrm{FM}}(x_t, y_t, c_t)\big|_{w = w_0},$$

where $(x_t, y_t) \sim d_0$ and $c_t \sim \mathsf{Feeder}(\cdot \mid x_t, y_t)$ are i.i.d. across $t$. Then $\{s_t\}_{t=1}^T$ are i.i.d. with mean $m_{\mathrm{FM}}$ and centered covariance $C_{\mathrm{FM}}$.

By the update rule,

$$w_T - w_0 \ = \ \rho\lambda \sum_{t=1}^T s_t, \qquad \text{so} \qquad \Pi(w_T - w_0) \ = \ \rho\lambda \sum_{t=1}^T \Pi s_t.$$

Expand the squared norm:

$$\|\Pi(w_T - w_0)\|_2^2 = \rho^2 \lambda^2 \Big\| \sum_{t=1}^T \Pi s_t \Big\|_2^2 = \rho^2 \lambda^2 \sum_{i=1}^T \sum_{j=1}^T \langle \Pi s_i, \Pi s_j \rangle.$$

Take expectation and use independence. For $i \ne j$,

$$\mathbb{E}\langle \Pi s_i, \Pi s_j \rangle = \big\langle \mathbb{E}[\Pi s_i], \, \mathbb{E}[\Pi s_j] \big\rangle = \|\Pi m_{\mathrm{FM}}\|_2^2.$$

For $i = j$, write $s_i = (s_i - m_{\mathrm{FM}}) + m_{\mathrm{FM}}$ and note

$$\mathbb{E}\|\Pi s_i\|_2^2 = \mathbb{E}\|\Pi(s_i - m_{\mathrm{FM}})\|_2^2 + \|\Pi m_{\mathrm{FM}}\|_2^2 = \mathrm{tr}\big(\Pi C_{\mathrm{FM}} \Pi\big) + \|\Pi m_{\mathrm{FM}}\|_2^2.$$

Therefore,

$$\mathbb{E}\big[\|\Pi(w_T - w_0)\|_2^2\big] = \rho^2 \lambda^2 \Big( T\big(\operatorname{tr}(\Pi C_{\mathrm{FM}}\Pi) + \|\Pi m_{\mathrm{FM}}\|_2^2\big) + T(T-1)\,\|\Pi m_{\mathrm{FM}}\|_2^2 \Big)$$
$$= \rho^2 \lambda^2 \Big( T\operatorname{tr}(\Pi C_{\mathrm{FM}}\Pi) + T^2\,\|\Pi m_{\mathrm{FM}}\|_2^2 \Big),$$

which proves the identity.

For the lower bound, Assumption D.3 gives $\Pi C_{\mathrm{FM}}\Pi \succeq \gamma_{\mathrm{FM}}\Pi$, hence

$$\operatorname{tr}(\Pi C_{\mathrm{FM}}\Pi) \ \geq \ \gamma_{\mathrm{FM}}\operatorname{tr}(\Pi) = \gamma_{\mathrm{FM}}\dim(S_{\mathrm{low}}(\eta)).$$

Assumption D.2 gives $\|\Pi m_{\mathrm{FM}}\|_2 \geq b_{\mathrm{FM}}$. Substituting these into the identity yields the stated bound. $\qquad\square$

# E. Experiment Details

## E.1. Prompts

---

**Feedback Provider Prompt**

You are an expert grader for math/logic problems.
Problem: {PROBLEM}
Student Solution: {LEARNER RESPONSE}
Your task:
- Analyze the student solution step by step.
- Focus on correctness and logical consistency.
- Identify potential mistake(s), if any.
- Provide concrete, actionable hints to improve the solution.
- Keep the Critique section under 200 words.

Format your response exactly as:
Thinking: [Your step-by-step analysis]
Critique: [Your final critique in under 200 words, ending with either "Your previous attempt was correct." or "Your previous attempt was incorrect."]

---

**Policy Prompt**

Question: {QUESTION}
You are given your previous attempt and an expert critique of it below. Your task is to produce an improved solution using the critique.
Your Previous Solution: {PREVIOUS RESPONSE}
Expert Critique: {TEXT FEEDBACK}
Instructions:
- Write your answer as a fresh solution to the original problem. Do not refer to your previous attempt.
- Do not mention or refer to the critique or the revision process.
- Use the critique only to improve correctness, clarity, and reasoning.
- Avoid using phrases like "Correctly applying the critique..." or "Reexamining my earlier solution...", etc., as the final answer should stand alone.

Let's think step by step and output the final answer within boxed{}.

---

## E.2. Case Studies

To demonstrate the purpose of text feedback, we study some example generations from the Shortest Path task. The goal is to output a sequence of moves (e.g., "right, down, right") from the start (*) to the destination (#), without passing through the obstacles (X). First, we show below an example from earlier in training (step 156), where the first attempt is incorrect and the second attempt is correct. The model incorrectly states that it is infeasible to get to the destination. The expert critique localizes the error by pointing out that moving down to (1,1) is valid. In the second attempt, the model incorporates this feedback and correctly outputs the sequence "right, down, down, down." (Interestingly, we observed that in regular single-turn GRPO, the model gets stuck in the local optimum of outputting "infeasible" regardless of the input; text feedback is critical for escaping this pitfall). Second, we show an example from the end of training (step 616), to show that the model has incorporated improved reasoning skills into even its first attempt capabilities.

---

**Shortest Path Example (Step 123)**

Question:
Find the shortest path from the start (*) to the destination (#) in this grid. You can only move through open cells (O) and cannot pass through obstacles (X). Valid moves are: right, down, left, up.
Grid:
X O X X O O

---

O X O O X X
X # O X O O
O O O O X X
X X O X X O
O X * O O X

If a path exists, return it as a sequence of moves (e.g., "right, down, right"). If no path exists, return "infeasible". Let's think step by step and output the final answer within boxed{}.

First Attempt:
Step 1: Identify the start and destination cells in the grid
The start (*) is located at row 5, column 5, and the destination (#) is at row 2, column 2.
Step 2: Examine the immediate surroundings of the start cell
The start cell at row 5, column 5 is blocked by an obstacle (X) to the north.
Step 3: Investigate possible movements from the start cell
There is no valid movement to the left or up from the start cell due to obstacles (X).
Step 4: Determine the feasibility of reaching the destination
Given that the start cell at row 5, column 5 is blocked by an obstacle to the north, and there are no valid movements to the left or up, it is not possible to reach the destination (#) from the start (*).
Step 5: Conclusion on the path from start to destination
The analysis reveals that the start cell is blocked by an obstacle and there are no valid movements to the left or up, making it infeasible to reach the destination (#) from the start (*).
The final answer is: infeasible

Expert Critique:
Your previous attempt was incorrect. You misidentified the start and destination coordinates. Double-check the grid by counting rows and columns carefully, starting from the top-left as (0,0). The start (*) is at row 5, column 2 (0-indexed), and the destination (#) is at row 2, column 1. From the start, you can move up to (4,2), which is open. From there, continue exploring valid paths. Re-examine the grid using the correct positions and trace possible moves step by step. Focus on building a path using only open cells (O), avoiding obstacles (X).

Second Attempt:
Step 1: Correctly identify the start and destination coordinates
The start (*) is at row 5, column 2 (0-indexed), and the destination (#) is at row 2, column 1.
Step 2: Analyze possible movements from the start cell
From the start at (5,2), move up to (4,2), which is an open cell (O).
Step 3: Continue exploring valid paths
From (4,2), we can move up to (3,2), which is also an open cell (O).
Step 4: Further explore valid paths from (3,2)
Moving up from (3,2) takes us to (2,2), another open cell (O).
Step 5: Determine the feasibility of reaching the destination
The destination (#) at (2,1) is reachable by moving left from (2,2).
Step 6: Conclusion on the path from start to destination
It is possible to reach the destination (#) from the start (*) by moving up from (5,2) to (4,2), then to (3,2), and finally to (2,2), followed by a valid left move to (2,1).
The final answer is: up, up, up, left

---

## Shortest Path Example (Step 616)

Question:
Find the shortest path from the start (*) to the destination (#) in this grid. You can only move through open cells (O) and cannot pass through obstacles (X). Valid moves are: right, down, left, up.
Grid:
O X X O O O
X O X O O #

X * X X O X
X O O O O O
O O X O X X
O O O O X O

If a path exists, return it as a sequence of moves (e.g., "right, down, right"). If no path exists, return "infeasible". Let's think step by step and output the final answer within boxed{}.

First Attempt:
Step 1: Identify the Start Location
The start (*) is located at row 2, column 1 on the grid.
Step 2: Identify the Destination
The destination (#) is located at row 1, column 5 on the grid.
Step 3: Explore the Grid from the Start
From row 2, column 1, we begin by examining the cell directly to the right. We find that row 2, column 2 is an X and cannot be visited. We continue by examining the cell directly below row 2, column 1. We find that row 3, column 1 is an O and can be visited. From row 3, column 1, we continue by examining the cell directly to the right. We find that row 3, column 2 is an O and can be visited. From row 3, column 2, we continue by examining the cell directly to the right. We find that row 3, column 3 is an O and can be visited. From row 3, column 3, we continue by examining the cell directly to the right. We find that row 3, column 4 is an O and can be visited. From row 3, column 4, we continue by examining the cell directly above. We find that row 2, column 4 is an O and can be visited. From row 2, column 4, we continue by examining the cell directly above. We find that row 1, column 4 is an O and can be visited. From row 1, column 4, we continue by examining the cell directly to the right. We find that row 1, column 5 is # and is the destination.
Step 4: Conclusion
We conclude that the path from the start (*) at row 2, column 1 involves continuing by examining the cell directly below to row 3, column 1, followed by row 3, column 2, row 3, column 3, row 3, column 4, row 2, column 4, row 1, column 4, and finally row 1, column 5.

Below, we also show example generations from `Knights and Knaves` and `DeepMath`.

---

**Knights and Knaves Example (Step 89)**

Question:
A very special island is inhabited only by sages and fools. Sages always tell the truth, and fools always lie. You meet 5 inhabitants: Lucas, Avery, Sebastian, Emily, and Elizabeth. "it is not the case that (Avery is a sage and Emily is a sage)," Lucas mentioned. "Lucas is a sage," Avery declared. Sebastian said, "if (Sebastian is a sage or Elizabeth is a fool) then (Avery is a sage and Sebastian is a sage)." "(if Lucas is a fool then Emily is a sage) if and only if Sebastian is a sage," Emily declared. "(Sebastian is a fool or Elizabeth is a sage or Emily is a fool) and (Sebastian is a fool if and only if Lucas is a fool) and Avery is a sage," Elizabeth declared. So who is a sage and who is a fool? (Format your answer like: "Lucas is a sage/fool, Avery is a sage/fool, Sebastian is a sage/fool, Emily is a sage/fool, and Elizabeth is a sage/fool") Let's think step by step and output the final answer within boxed{}.

First Attempt:
Step 1: Analyze Lucas's statement Lucas said, "it is not the case that (Avery is a sage and Emily is a sage)." This statement implies that either Avery is not a sage or Emily is not a sage (or both).
Step 2: Analyze Avery's statement Avery declared, "Lucas is a sage." Since sages always tell the truth, Avery must be a sage because he is correctly identifying Lucas as a sage.
Step 3: Analyze Sebastian's statement Sebastian said, "if (Sebastian is a sage or Elizabeth is a fool) then (Avery is a sage and Sebastian is a sage)." Given that Avery is a sage from Avery's statement, for Sebastian's conditional statement to hold true (and be truthfully revealed by a sage who is a side of the or condition), Sebastian must also be a sage (since being a fool would require him to lie).
Step 4: Analyze Emily's statement Emily declared, "(if Lucas is a fool then Emily is a sage) if and only if Sebastian is a sage." Since Sebastian is a sage from step 3, the if and only if part is relevant. Given that Sebastian is a sage, for the statement to be true (as it is made by a sage), the premise about Lucas being a fool implying Emily is a sage

must be false. Thus, Lucas is not a fool.

Step 5: Analyze Elizabeth's statement Elizabeth declared, "(Sebastian is a fool or Elizabeth is a sage or Emily is a fool) and (Sebastian is a fool if and only if Lucas is a fool) and Avery is a sage." Given that Avery is indeed a sage from step 2, the final part of the statement is confirmed to be true. The first part, with all the conditions, would require at least one of the conditions involving Sebastian or Elizabeth or Emily being a fool, but since we've established Lucas is not a fool, and Sebastian is a sage from step 3, the statement holds true for any of these being a fool. However, given all the previous deductions, the direct implication from this statement doesn't necessarily reveal new truth but solidifies existing truths.

Step 6: Summarize the deductions - Lucas is a sage because he's not presented as a liar and his statement about Avery and Emily is factually considered in relation to other statements. - Avery is a sage, confirmed by the truthfulness of his statement. - Sebastian is a sage, deduced from the conditional statement and its implications on his own truthfulness. - Emily is a sage because her statement about the relation of Sebastian being a sage and the implication on Lucas needing to be a fool for her to be a sage is not relevant given Sebastian's truth status. - Elizabeth is a fool because her statement, given as a sage, requires conditions that could be fulfilled by lies within her own statement, and given the deductions made are based on the truthfulness of others. The final answer is: boxed{Lucas is a sage, Avery is a sage, Sebastian is a sage, Emily is a sage, and Elizabeth is a fool}

Expert Critique:

Your previous attempt was incorrect. The major error is assuming Avery is a sage without proof—this is circular. You cannot assume Avery is a sage just because he says Lucas is one. Similarly, assuming Sebastian is a sage based on the truth of the consequent is invalid—remember that the truth of a conditional depends on both truth value and structure. The biconditional in Emily's statement must be evaluated more carefully: if Sebastian is a fool, the biconditional requires the implication (if Lucas is fool → Emily is sage) to be false. Try testing assumptions with truth tables or counterfactuals. Identify which statements force contradictions if certain roles are assumed. Focus on Elizabeth's statement—it's long but has a "and" structure, so for a fool, *at least one* part must be false. This can help eliminate possibilities.

Second Attempt:

Step 1: Initial Analysis We begin by analyzing the statements given by each inhabitant of the island, remembering that sages always tell the truth, and fools always lie. Our goal is to determine who is a sage and who is a fool.

Step 2: Analyzing Lucas's Statement Lucas says, "it is not the case that (Avery is a sage and Emily is a sage)." This statement implies that either Avery is not a sage or Emily is not a sage (or both).

Step 3: Analyzing Avery's Statement Avery declares, "Lucas is a sage." To determine if Avery is a sage or a fool, we must evaluate his statement in the light of Lucas's character. We cannot immediately assume Avery is a sage because that would be based on his own claim, which may or may not be true.

Step 4: Analyzing Sebastian's Statement Sebastian says, "if (Sebastian is a sage or Elizabeth is a fool) then (Avery is a sage and Sebastian is a sage)." This conditional statement necessitates a closer look. The antecedent (Sebastian is a sage or Elizabeth is a fool) cannot be evaluated immediately because we need to determine Sebastian's character.

Step 5: Analyzing Emily's Statement Emily says, "(if Lucas is a fool then Emily is a sage) if and only if Sebastian is a sage." This statement involves a biconditional and must be evaluated carefully. If Sebastian were a fool, for Emily to be speaking the truth, the implication (if Lucas is fool → Emily is sage) must be false, but since we cannot determine Sebastian's character yet, we must proceed with caution.

Step 6: Analyzing Elizabeth's Statement Elizabeth says, "(Sebastian is a fool or Elizabeth is a sage or Emily is a fool) and (Sebastian is a fool if and only if Lucas is a fool) and Avery is a sage." Given the "and" structure, for Elizabeth to be a fool, at least one part of her statement must be false. However, we need to see how Avery's truthfulness affects this analysis.

Step 7: Evaluating Truth Conditions We must evaluate each statement under the condition that the speaker might be either a sage or a fool. Lucas's statement does not depend on the truth of others in a direct linear fashion but rather sets the scene for the entire puzzle. Avery's statement can only be considered under the separation of Lucas being a sage or not due to the nature of the puzzle's setup.

Step 8: Truth Table Approach Given the complexity of the statements, a truth table approach could help. However, without specific conditions or values assigned to Lucas, Avery, Sebastian, Emily, and Elizabeth, and without the ability to directly evaluate their truthfulness based on their statements alone without circular reasoning, we must focus on the logical structure of the statements they made.

Step 9: Focusing on Logical Implications We should identify which statements, if assumed, would force contradictions. Key implications include Elizabeth's statement having two distinct components, one of which involves the relationship between Sebastian and Lucas and another related to their characters and the truth of Avery's sagehood.

Step 10: Identifying Constraints Given the structure of the statements, if we can establish that there is at least one condition (from Elizabeth's statement) that Elizabeth, as a potential fool, must have false, this could give us a concrete path towards determining who is a sage and who is a fool, considering the truthfulness of the other inhabitants.

Step 11: Contradiction Test Let's directly test for contradictions by making assumptions about who is a sage or a fool and determining if each inhabitant's statement would hold true or not under those assumptions.

Step 12: Establishing Truth and Falsehood Through process of elimination and evaluating the logical implications of the statements, especially considering the "and" structure of Elizabeth's statement, we can establish who must be a sage or a fool.

Step 13: Finalizing Roles Based on the logical analysis and the conditions set forth by each inhabitant, we can assign roles of sage or fool to each of the individuals.

Step 14: Conclusion After analyzing the statements and their logical implications, we can now determine who is a sage and who is a fool among the inhabitants of the island. The final answer is: boxed{Lucas is a sage, Avery is a sage, Sebastian is a fool, Emily is a fool, and Elizabeth is a fool}

---

## Deepmath Example (Step 1290)

Question:
Let $x_1, x_2, \ldots, x_{1970}$ be positive integers such that $x_1 + x_2 + \cdots + x_{1970} = 2007$. Find the largest possible value of $x_1^3 + x_2^3 + \cdots + x_{1970}^3$. Let's think step by step and output the final answer within boxed{}.

First Attempt:
Step 1: Understand the problem We are given a sequence of 1970 positive integers, $x_1, x_2, \ldots, x_{1970}$, such that their sum is 2007. We need to find the largest possible value of the sum of the cubes of these integers, i.e., $x_1^3 + x_2^3 + \cdots + x_{1970}^3$.

Step 2: Use the concept of weighted inequalities A common approach to problems of this type is to use inequalities that relate the sum of powers to the sum of the numbers, such as Cauchy-Schwarz or inequalities involving the sum of cubes. One powerful tool is Cauchy-Schwarz, but here we might consider inequalities that compare sums of integers to sums of their cubes, such as $\sum x_i^3 \geq \frac{(\sum x_i)^3}{1970}$, although we must carefully consider how to apply such inequalities and whether they give the maximum.

Step 3: Recall the power mean inequality The power mean inequality can be useful here, particularly the inequality that compares the arithmetic mean and the cubic mean. However, our main concern is maximizing $\sum x_i^3$, subject to $\sum x_i = 2007$. A more direct approach involves focusing on how to distribute the sum of 2007 across the 1970 integers to maximize the total cubic sum.

Step 4: Consider the equality case for weighted sums For inequalities that relate sums of powers, such as $\sum x_i^3 \geq \frac{(\sum x_i)^3}{n}$, equality occurs when all $x_i$ are equal. However, since we are looking to maximize this sum under a constraint and the numbers must be positive integers, we cannot set all $x_i = 1$, as that would sum to 1970, not 2007. Instead, we aim to allocate as much of the sum as possible to the largest $x_i$, because $x^3$ grows faster than $x$, so concentrating the sum in the largest possible value will generally increase the cube sum more than spreading it out. [Skip parts of irrelevant generation]

Step 29: Evaluate the cube of $x_i = 1$ If all $x_i = 1$, then $\sum x_i^3 = 1970 \cdot 1^3 = 1970$.

Step 30: Use the fact that the sum of cubes is maximized by making one $x_i$ as large as possible We seek to make one $x_i$ as large as possible while ensuring all are positive integers and the sum is 2007. If we set all but one $x_i = 1$, that leaves 2007 - 1969 = 38 to be assigned to the remaining $x_i$. So, we could set one $x_i = 38$ and the rest = 1.

Step 31: Calculate the sum of cubes for $x_i = 1$ and one $x_i = 38$ If one $x_i = 38$ and the rest = 1, then the sum of cubes is $1^3 + 1^3 + \cdots + 1^3 + 38^3$, which is $1969 \cdot 1^3 + 38^3$.

Step 32: Perform the calculation of $38^3$ $38^3 = 38 \times 38 \times 38 = 54,872$.

Step 33: Add 1969 to the result of $38^3$ $54,872 + 1969 = 57,841$.

Step 34: Evaluate the total sum of cubes The total sum of cubes is $1969 + 57,841 = 59,810$.

Step 35: Compare this to the sum when all $x_i = 1$ When all $x_i = 1$, the sum of cubes is 1970. Our calculated sum is 59,810, which is larger.

Step 36: Confirm the maximum Based on the calculation, the maximum possible value of $\sum x_i^3$ under the given constraints is 59,810. The final answer is: $\boxed{59110}$

Expert Critique:

Your approach to maximize the sum of cubes by concentrating value in one variable is correct and well-reasoned. However, you made an arithmetic error when computing the total sum of cubes: $54,872 + 1,969 = 56,841$, not 59,110. Double-check your additions, especially when combining large numbers. Your allocation of one 38 and 1969 ones is valid and optimal under the constraints. The error is purely computational. Recheck the sum: $38^3 = 54,872$, plus 1969 ones contributes 1,969. Add them carefully. Your previous attempt was incorrect.

Second Attempt:

Step 1: Understand the problem We are given 1970 positive integers, denoted as $x_1, x_2, \ldots, x_{1970}$, with the constraint that their sum equals 2007. Our goal is to find the largest possible value of the sum of their cubes, i.e., $x_1^3 + x_2^3 + \cdots + x_{1970}^3$.

Step 2: Analyze the constraint and objective The constraint is $x_1 + x_2 + \cdots + x_{1970} = 2007$, and we want to maximize $x_1^3 + x_2^3 + \cdots + x_{1970}^3$. Since the cubic function $f(x) = x^3$ is strictly increasing for positive $x$, the maximum of the sum of cubes will occur when the values are concentrated in as few variables as possible, with the largest possible value assigned to one variable.

Step 3: Reason about the distribution To maximize the sum of cubes, we should make one integer as large as possible and the rest as small as possible, while still satisfying the sum constraint. This is because the cube function grows very rapidly, so even a small increase in one variable can lead to a much larger increase in its cube.

Step 4: Calculate the maximum possible value We want to assign the largest possible value to one variable, say $x_1$. The maximum possible value for $x_1$ occurs when we set all other variables to the smallest possible value, which is 1. The sum constraint tells us that $x_1 + 1969 \cdot 1 = 2007$, so we solve for $x_1$: $x_1 = 2007 - 1969 = 38$.

Step 5: Calculate the sum of cubes Now we calculate the sum of cubes under this allocation: $x_1^3 + x_2^3 + \cdots + x_{1970}^3 = 38^3 + 1969 \cdot 1^3$. First, compute $38^3 = 38 \times 38 \times 38 = 54,872$. Then, multiply $1969 \times 1^3 = 1969$. Finally, add these two values together.

Step 6: Perform the addition carefully Now we add $54,872 + 1969$ carefully. Starting with $54,000 + 1969 = 55,969$, then add the remaining 892 to get $56,861$. No, we must do it precisely: $54,872 + 1,969 = 56,841$.

Step 7: Confirm the optimal allocation We have found that the optimal allocation is to set one variable to 38 and the rest to 1. This is valid under the constraints, and the sum of cubes under this allocation is $56,841$.

Step 8: Conclude the maximum sum of cubes Therefore, the largest possible value of the sum of cubes $x_1^3 + x_2^3 + \cdots + x_{1970}^3$ is $56,841$. The final answer is: $\boxed{56841}$.

### E.3. Details of Benchmarks

In this subsection, we describe our evaluation setup for each domain.

**Reasoning.** Reasoning Gym (Stojanovski et al., 2025) is a library of reasoning puzzles spanning categories such as algebra, arithmetic, games, geometry, graphs, induction, and logic. We evaluate on three tasks from Reasoning Gym: `Knights and Knaves`, `Binary Matrix`, and `Shortest Path`. `Knights and Knaves` is a logic puzzle where some characters are truthful "knights" and the rest are untruthful "knaves", and the goal is to deduce from the characters' statements whether they are knights or knaves. `Binary Matrix` is an algorithmic problem, where the input is a square matrix of 1's and 0's and the task is to find the Manhattan distance to the nearest 0 for each cell. `Shortest Path` is a graph problem, where the input is a matrix containing *, #, O, and X, representing the start cell, the destination cell, an open cell, and a blocked cell, respectively. The objective is to find the shortest path from the start to the destination, using only open cells. For all environments, we procedurally generate 20K problems in similar style to the Reasoning Gym code [4]. We use 19.8K examples for training and the remaining 200 examples for testing.

**Math.** We evaluate on two standard math benchmarks, MATH500 and AIME24. MATH500 (Hendrycks et al., 2021) is a competition math dataset of 500 problems spanning seven categories: algebra, counting and probability, geometry, intermediate algebra, number theory, prealgebra and precalculus. The American Invitational Mathematics Examination (AIME) is a prestigious high school math competition. There are two version of the exam with 15 questions each, for a total of 30 questions in the 2024 test set. We consider training on two training sets: DAPO-17K (Yu et al., 2025), which contains 17K problems sourced from math competitions, and DeepMath-103K (He et al., 2025), which contains 103K problems that are of generally higher difficulty than DAPO-17K.

**Creative writing.** LitBench (Fein et al., 2025) is a dataset containing 43K pairwise examples of LLM-generated stories and human preference labels. We train on the LitBench train set and evaluate on the test set of 2K examples. We also use the same checkpoint and evaluate on WritingBench (Wu et al., 2025), which is a benchmark containing 1K real-world writing tasks, spanning academics and engineering, finance and business, politics and law, literature and art, education, and advertising and marketing. For evaluation, we use the prompt-specific rubric from WritingBench and GPT-4.1-mini as the judge.

---

[4] https://github.com/open-thought/reasoning-gym

### E.4. Hyperparameters

Hyperparameters specific to Reasoning Gym environment configurations:

- Knights and Knaves: `n_people=5, depth_constraint=3, width_constraint=3`
- Binary Matrix: `min_n=3, max_n=5`
- Shortest Path: `min_rows=5, min_cols=5, max_rows=6, max_cols=6, p_blocked=0.4`

*Table 2.* Hyperparameters for all multi-turn RL experiments.

| Hyperparameter | Value |
|---|---|
| Group size | 8 |
| Groups per batch | 32 |
| Max tokens | 8096 |
| LoRA rank | 32 |
| Learning rate | 2e-5 |
| KL penalty | 0.0 |
| RL coefficient (Self Distillation) | 0.1 |
| SFT coefficient (Feedback Modeling) | 0.2 |

For math training, we use a linear schedule for the RL coefficient from 0 to 0.1.

**Feedback Descent.** There is no publicly available codebase for Feedback Descent, so we reimplement the algorithm based on details from the Feedback Descent (Lee et al., 2025) paper and the GEPA (Agrawal et al., 2025) paper. For the candidate comparison prompt, we use "System Prompt Template for Prompt Optimization" from the Feedback Descent paper. Across all tasks, we perform 5 rounds of prompt improvement on the training set, collect all candidate prompts encountered during optimization, evaluate them on a held-out validation set, and select the best-performing prompt for final evaluation on the test set. We used a temperature of 0.6 for Reasoning Gym and competition math and a temperature of 0.7 for creative writing. For Reasoning Gym tasks, we use 200 training and 200 validation examples. For the competition math setups, we use 64 training and 16 validation examples. For creative writing, we use 20 training and 16 validation examples. We perform evaluations on the full test set. While these training sets are smaller than the fully available training data, we aimed to follow the precedent of prior prompt optimization work (e.g., GEPA uses on the order of 100-150 examples for training) while using a reasonable amount of compute for our budget. Prompt optimization is generally highly sample-efficient, and we also observed empirically that increasing the number of training and validation examples did not yield meaningful performance improvements.

### E.5. Variance of Importance Weighting

Let $q(\cdot)$ denote the second-turn (teacher) distribution over outputs, induced by one round of feedback: $y_1 \sim q(\cdot) := \pi(\cdot \mid x_1)$ with $x_1 = f(x_0, y_0, c_0)$. Recall that an unbiased estimator of the single-turn policy gradient can be obtained by importance weighting with $w(y) = \pi(y \mid x_0)/q(y)$, but its variance is controlled by the second moment $\mathbb{E}_q[w^2]$. In particular, $\mathbb{E}_q[w^2] = 1 + \chi^2(\pi(\cdot \mid x_0)\|q) = \exp(D_2(\pi(\cdot \mid x_0)\|q))$ [5], thus unlike the commonly used KL divergence, the variance is measured with directly density ratio instead of the logarithm of the density ratios.

To measure the variance empirically, we consider the AIME24 task, and we measure the average trajectory-level log ratio and average token-level log ratio between the first-turn and second-turn policies, and we summarize the results in Table 3. Note that the trajectory-level log ratio is almost vacuously large (recall that the variance if measure on the direct policy ratio, thus the exponential of the log ratio), indicating that importance weighting at the trajectory level is infeasible. On the other hand, the token-level log ratio also exhibits high variance, indicating long tail behavior. This explains the difficulty of applying importance weighting in practice.

*Table 3.* Average log importance weight ratios between first-turn and second-turn policies on AIME24.

|                               | Value     |
| ----------------------------- | --------- |
| Average (Trajectory)          | 142.1637  |
| Average (Token)               | 0.1288    |
| Standard Deviation (Token)    | 0.5506    |

---

[5]For probability measures $p$ and $q$ with $p \ll q$, the $\chi^2$-divergence and order-2 Rényi divergence are $\chi^2(p\|q) = \int \frac{(p(y)-q(y))^2}{q(y)} \, dy$, and $D_2(p\|q) = \log \int \frac{p(y)^2}{q(y)} \, dy$.

*Table 4.* Comparison of baselines across **reasoning puzzles**, **competition math**, and **creative writing** tasks. We report second-turn accuracy after 2-turn training (i.e., $J_{\text{SingleTurn}}(\pi)$) of the last checkpoint. For the reasoning tasks and LitBench, we report the mean@1 accuracy, judged by either verifiable reward or LLM-as-a-judge. For the math tasks, we report the mean@32 accuracy from the last checkpoint from the training. The parentheses denote the training dataset. For WritingBench, we follow the official protocol with GPT-4.1-mini as the judge. The accuracy in reasoning and math is normalized between 0 and 1, and the score in creative writing is normalized between 1 and 10. Note that Self Distillation and Feedback Modeling consistently outperform all baselines across tasks.

| | Base Model | GRPO 2-turn | RLTF-SD | RLTF-FM |
|---|---|---|---|---|
| **Reasoning** | | | | |
| Knights and Knaves | 0.119 | 0.636 | 0.896 | 0.910 |
| Binary Matrix | 0.001 | 0.978 | 0.989 | 0.993 |
| Shortest Path | 0.034 | 0.836 | 0.892 | 0.976 |
| **Math** | | | | |
| Math500 (DAPO) | 0.583 | 0.692 | 0.725 | 0.771 |
| AIME24 (DAPO) | 0.042 | 0.142 | 0.185 | 0.250 |
| Math500 (Deepmath) | 0.583 | 0.741 | 0.765 | 0.793 |
| AIME24 (Deepmath) | 0.042 | 0.133 | 0.203 | 0.208 |
| **Creative writing** | | | | |
| LitBench | 5.84 | 8.02 | 9.20 | 8.60 |

## E.6. Additional Results

Here, we provide evaluation curves for our general results in reasoning, math and creative writing, corresponding to Tables 1 and 4. Figure 5 shows the single-turn accuracy and Figure 6 shows the multi-turn accuracy.

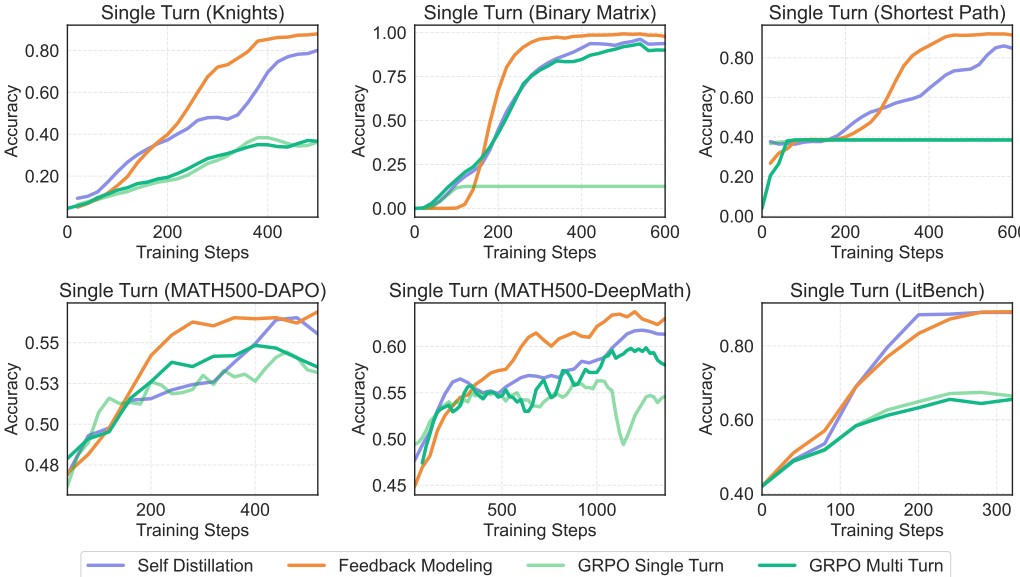

*Figure 5.* Evaluation curves for single-turn accuracy across **reasoning puzzles**, **competition math**, and **creative writing** tasks, at every 40 training steps. For reasoning tasks, we report the mean@1 accuracy judged by either verifiable reward or LLM-as-a-judge. For math tasks, we report the mean@1 accuracy. For Litbench, we report the mean@1 accuracy judged by LLM-as-a-judge. The accuracy in reasoning and math is normalized between 0 and 1, and the score in creative writing is normalized between 1 and 10.

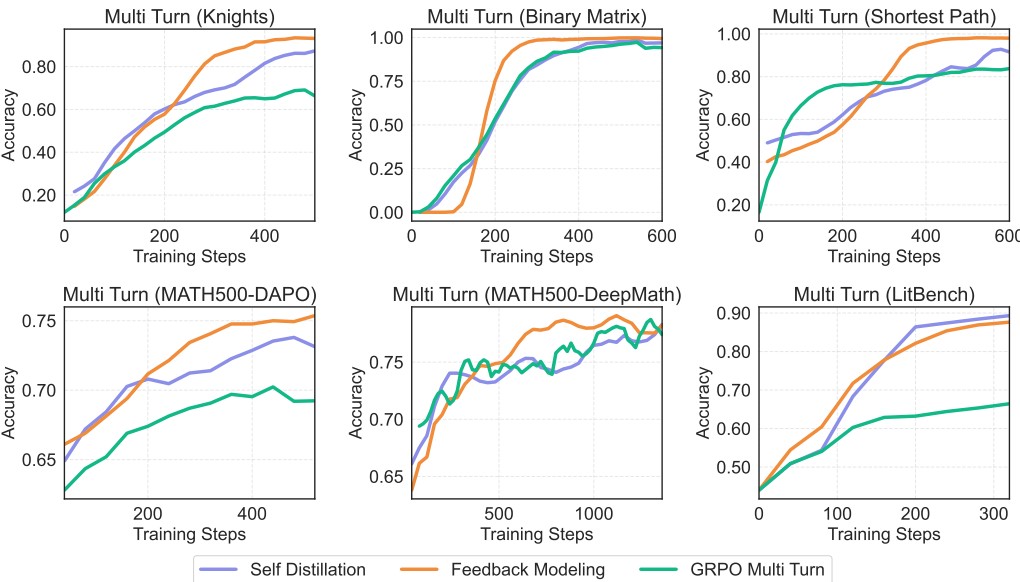

*Figure 6.* Evaluation curves for multi-turn accuracy across **reasoning puzzles**, **competition math**, and **creative writing** tasks, at every 40 training steps. For reasoning tasks, we report the mean@1 accuracy judged by either verifiable reward or LLM-as-a-judge. For math tasks, we report the mean@1 accuracy. For `Litbench`, we report the mean@1 accuracy judged by LLM-as-a-judge. The accuracy in reasoning and math is normalized between 0 and 1, and the score in creative writing is normalized between 1 and 10.

## E.7. Resources

All of our training experiments are performed using Tinker (Lab, 2025).

