# OpenReview forum: "Expanding the Capabilities of Reinforcement Learning via Text Feedback"
_ICML.cc/2026/Conference — ICML 2026 regular_

### Official Review · Reviewer_X7ok · 2026-02-20

**Soundness:** 3
**Presentation:** 4
**Significance:** 3
**Originality:** 3
**Overall Recommendation:** 5
**Confidence:** 3

**Summary:**

This paper tackles the limitation of RL post-training for LLMs where the supervision signal is often a sparse scalar reward. A reward of 0 provides little information about why an output is wrong, making learning inefficient when the base model struggles. The authors propose leveraging natural language critiques as an intermediate supervision signal. They formalize a multi-turn training setup where the model produces an answer, receives critique, and revises, but evaluation focuses on improving single-turn performance where critique is unavailable at test time.

Training uses two sources of signal:

1. Standard RL objective (scalar reward: verifiable for reasoning/math, judge-based for writing).

2. Feedback-based auxiliary learning, via either:

-  Self Distillation: distill the model’s own critique-conditioned second-turn response into the first-turn policy

- Feedback Modeling: train the model to predict the critique text as an auxiliary loss.

They evaluate across reasoning, math, and creative writing benchmarks and include extensive ablations.

**Compliance With Llm Reviewing Policy:**

Affirmed.

**Final Justification:**

This paper studies how to improve LLM post-training with natural language feedback rather than relying solely on sparse scalar rewards, and one of its main strengths is how effectively it leverages this richer supervision signal. My main reservation was the reliance on a very large judge model, which I still view as a meaningful limitation. That said, I found the paper interesting and ultimately worthy of acceptance, as the authors provided convincing rebuttal evidence that the approach remains beneficial even with weaker judges. The added cross-model generalization experiments further increased my confidence. Overall, the rebuttal addressed my central concerns and strengthened my positive assessment.

**Key Questions For Authors:**

see weaknesses

**Limitations:**

yes

**Strengths And Weaknesses:**

**Strengths**

(1) The paper is clearly written, the motivation is easy to follow.

(2) The framing of training-time feedback, test-time single-turn performance is well-motivated and nontrivial.

(3) The Self Distillation design is thoughtful, in particular, the discussion of second-turn mean baselines and “gradient-signal collapse” is insightful and practically relevant.

(4) Results are strong across multiple domains, and the paper includes meaningful ablations isolating key choices.

(5) The qualitative examples in the appendix help interpret what kinds of errors critiques correct and how improvements propagate to the first turn.

**Weaknesses**

(1) Experiments use Qwen3-235B as the feedback provider for training a Llama-3.1-8B learner. Since the paper shows correctness-only feedback is much weaker, the effectiveness may rely heavily on the judge’s capability. An ablation on judge size would strengthen the claim that the method is broadly useful. It would also be valuable to discuss alternatives such as a learned judge (e.g., RL-Tango [1]).

(2) Given access to a 235B model, a natural baseline is to generate (i) improved second-turn solutions or (ii) full solutions/reasoning traces with the judge and then perform standard SFT/distillation on the student (possibly with filtering by verifiable reward). Without this, it is hard to disentangle gains from the proposed RL-from-feedback objectives vs. simply distilling from a strong teacher, since we have an access to it anyway.

(3) All results use the same learner (Llama-3.1-8B) and the same judge family (Qwen3-235B). It remains unclear whether gains persist with different learners or different judges (smaller or different families). Even one additional pairing would improve confidence.


**References**

[1] Zha, Kaiwen, et al. "Rl tango: Reinforcing generator and verifier together for language reasoning." arXiv preprint arXiv:2505.15034 (2025).

---

> ### Author Rebuttal · Authors · 2026-03-31
>
> We thank the reviewer for the careful and constructive review. We are glad the reviewer found the paper clearly written, the framing well-motivated, and the Self Distillation design (including gradient-signal collapse analysis) insightful. We address each concern below.
> >*An ablation on judge size would strengthen the claim that the method is broadly useful. It would also be valuable to discuss alternatives such as a learned judge (e.g., RL-Tango [1]).*
>
> We have added experiments using a smaller, less capable model as the judge, simulating the kind of noisy, imperfect feedback one would expect from non-expert human annotators or lightweight automated graders in practice. We show the results on knights and knaves (with the same learner model) below:
>
> |Judge Model|Self Distillation|Feedback Modeling|GRPO|
> |-|-|-|-|
> |Qwen3-235B-A22B-Instruct (in the paper)|0.802|0.880|0.352|
> |Qwen3-30B-A3B-Instruct|0.661|0.750|-|
> |Qwen3-4B-Instruct|0.609|0.758|-|
>
> We observe that both of **our methods (especially feedback modeling) are robust to the weaker feedback providers** (we use the model from the same model family to show a clean hierarchy), as the performance still greatly improves upon GRPO even with a much weaker model to simulate the feedback provider.
>
> Regarding RL-Tango: we thank the reviewer for the reference. RL-Tango studies jointly training a generator and verifier, which is complementary to our setting where the feedback provider is external. We will include a discussion in the revised paper.
>
> >*A natural baseline is to generate (i) improved second-turn solutions or (ii) full solutions/reasoning traces with the judge and then perform standard SFT/distillation*
>
> We want to clarify that **Qwen3-235B is not used as a teacher, it is used to simulate a feedback provider**. That is, the larger model is part of the learning environment, so we can not fully control the generation of the larger models (e.g., generating demonstrations or obtaining logits for distillation).
>
> Concretely, given the student's attempt, it produces a natural language critique identifying errors and suggesting improvements. It never generates correct solutions or reasoning traces. This mirrors how feedback works in practice: a compiler returns error messages, a unit test reports which cases fail, and a human marks what went wrong during a chat; none of these produce demonstrations. The 235B model is simply a convenient stand-in for these real-world feedback sources in a controlled experimental setting. Distilling from it would be analogous to distilling from a unit test framework: it conflates the feedback provider with a solution provider, which is not the setting we wish to model in our paper. Another analogy one can draw is in RLHF/alignment; sometimes people use LLM-as-a-judge to simulate human preference [1,2], but distilling from the judge LLM is not a fair baseline.
>
> That said, the reviewer's suggestion could help contextualize the gap between learning from feedback and learning from demonstrations. We show the comparison below:
>
> |  |GRPO|Self Distillation|Feedback Modeling|Distillation|Qwen3-235B-A22B-Instruct|
> |-|-|-|-|-|-|
> | Knights and Knaves |0.352|0.802|0.880|0.862|0.984|
>
> We observe that distillation performs slightly better than self-distillation, and feedback modeling actually achieves the same performance as distillation. While we do not believe this result can generalize to arbitrary setting, we believe this is a very positive result showing RLTF in some cases can be as performative as distillation, which is way more expensive and less practical.
>
> >*All results use the same learner (Llama-3.1-8B) and the same judge family (Qwen3-235B)...Even one additional pairing would improve confidence.*
>
> We agree that showing generalization across learner–judge pairings is important. We have added experiments using Qwen3-30B-A3B-Instruct as the learner, and GPT-OSS-120B to simulate the feedback provider. Qwen3-30B-A3B-Instruct is the cheapest Qwen model on tinker and we also want to cover the case of MoE models. We choose GPT-OSS-120B so we can test robustness to arbitrary model combinations. We choose knight swap because the current reasoning tasks are easy enough for the student model (without any training). We summarize the results below:
>
> | |GRPO Multi Turn|Self Distillation|Feedback Modeling|
> |-|-|-|-|
> |Knight Swap|0.225|0.305 (+37.3%)|0.365 (+62.2%)|
>
> These results confirm that our methods generalize beyond the Llama family. We remark that we constructed the same number of prompts in knight swap as in other reasoning environment, but at the end of training (1 epoch), the GRPO performance has saturated but both of our methods are still climbing, so we expect *the gap will be even larger under additional training*, but that is beyond our current resource limit.
>
> [1] Zheng et al. (2023). "Judging LLM-as-a-Judge with MT-Bench and Chatbot Arena”.
>
> [2] Bai et al. (2022). “Constitutional AI: Harmlessness from AI Feedback”.

---

> > ### Author Rebuttal · Reviewer_X7ok · 2026-04-01
> >
> > I would like to thank the reviewers for their detailed rebuttal. I find the authors’ arguments convincing, and I am therefore raising my score.

---

> > > ### Author Response · Authors · 2026-04-03
> > >
> > > We are glad to receive the recognition from the reviewer, and we thank the reviewer for raising the score.

---

### Official Review · Reviewer_1UMj · 2026-02-22

**Soundness:** 3
**Presentation:** 2
**Significance:** 2
**Originality:** 3
**Overall Recommendation:** 4
**Confidence:** 3

**Summary:**

The paper propose combining text and rult-based reward to propose the performance of LLM RL training. Empirical results show the effectiveness of the methods. The authors also provide some theoratic provement.

**Compliance With Llm Reviewing Policy:**

Affirmed.

**Final Justification:**

most of the concerns are solved, score updated

**Key Questions For Authors:**

See the weakness, the most important concern is the generalization on different model and time cost.

**Limitations:**

it is unclear the cost of the methods.

**Strengths And Weaknesses:**

## Strengths

- Both the self-distillation show good performance compare to previous methods in benchmark from different domain.

- Self-distillation provide a good near-in-policy training method to improve the performance of RL training.

- The paper is easy to follow


## Weakness

- The experiments mainly focus on Llama-based model, which is suboptimial in GRPO-style RL training[1]. Could you add some results from non-llama model like gemma or qwen?

- Since you are using a larger model during the training, could you provide the performance of on-ploicy distillation which using the Qwen3-235B-A22B-Instruct-2507 as teacher model.

- Could you provide the training cost(time and memory requirment) compared to previous method, since your method looks like quite slow because some non-paralllel step.

[1]OctoThinker: Mid-training Incentivizes Reinforcement Learning Scaling

---

> ### Author Rebuttal · Authors · 2026-03-31
>
> We thank the reviewer for their time and feedback. We address each concern below.
> >*The experiments mainly focus on Llama-based model, which is suboptimial in GRPO-style RL training[1]. Could you add some results from non-llama model like gemma or qwen?*
>
> We thank the reviewer for this suggestion. We have added experiments using Qwen3-30B-A3B-Instruct as the learner, and GPT-OSS-120B to simulate the feedback provider. Qwen3-30B-A3B-Instruct is the cheapest Qwen model on tinker and we also want to cover the case of MoE models. We choose GPT-OSS-120B so we can test robustness to arbitrary model combinations. We choose knight swap because the current reasoning tasks are easy enough for the student model (without any training). We summarize the results below:
>
> |  | GRPO Multi Turn | Self Distillation | Feedback Modeling |
> |-|-|-|-|
> | Knight Swap | 0.225  | 0.305 (+37.3%)  | 0.365 (+62.2%)  |
>
> These results confirm that our methods generalize beyond the Llama family. We remark that we constructed the same number of prompts in knight swap as in other reasoning environment, but at the end of training (1 epoch), the GRPO performance has saturated but both of our methods are still climbing, so we expect *the gap will be even larger under additional training*, but that is beyond our current resource limit.
>
> >*Since you are using a larger model during the training, could you provide the performance of on-ploicy distillation which using the Qwen3-235B-A22B-Instruct as teacher model.*
>
> We want to clarify that **Qwen3-235B is not used as a teacher, it is used to simulate a feedback provider**. That is, the larger model is part of the learning environment, so we can not fully control the generation of the larger models (e.g., generating demonstrations or obtaining logits for distillation).
>
> Concretely, given the student's attempt, it produces a natural language critique identifying errors and suggesting improvements. It never generates correct solutions or reasoning traces. This mirrors how feedback works in practice: a compiler returns error messages, a unit test reports which cases fail, and a human marks what went wrong during a chat; none of these produce demonstrations. The 235B model is simply a convenient stand-in for these real-world feedback sources in a controlled experimental setting. Distilling from it would be analogous to distilling from a unit test framework: it conflates the feedback provider with a solution provider, which is not the setting we wish to model in our paper. Another analogy one can draw is in RLHF/alignment; sometimes people use LLM-as-a-judge to simulate human preference [2,3], but distilling from the judge LLM is not a fair baseline.
>
> That said, to contextualize the gap between learning from feedback and learning from demonstrations, we include an off-policy distillation baseline using Qwen3-235B as a teacher (on-policy distillation is not applicable as the teacher and student use different tokenizers, workarounds exist [1] but it is beyond the scope of this rebuttal):
> |  | GRPO | Self Distillation | Feedback Modeling | Distillation | Qwen3-235B-A22B-Instruct |
> |-|-|-|-|-|-|
> | Knights and Knaves | 0.352 | 0.802  | 0.880 | 0.862 | 0.984 |
>
> We observe that distillation performs slightly better than self-distillation, and feedback modeling actually achieves the same performance as distillation. While we do not believe this result can generalize to arbitrary setting, we believe this is a very positive result showing RLTF in some cases can be as performative as distillation, which is way more expensive and less practical.
>
> Finally, we also conduct experiments where we use a weaker model to simulate the feedback provider, and we think that might be complementary to the above results. Due to character limitation we refer the reviewer to the second table in our response to reviewer XNey.
>
> >*Could you provide the training cost(time and memory requirment) compared to previous method, since your method looks like quite slow because some non-paralllel step.*
>
> Our methods introduce negligible additional cost over multi-turn GRPO, which is a baseline in our experiments. The overhead is limited to computing one additional loss term on already-collected data, which is negligible relative to the cost of generation. To provide a concrete example, on knights and knaves, multi-turn GRPO took 38.5 hours, self-distillation took 42 hours, and feedback modeling took 44 hours, but note that these are subject to variability on the tinker side.
>
> More broadly, a key motivation of our work is that text feedback is already abundant in practice. Our methods extract more learning signal from this existing data at essentially zero marginal cost.
>
>
> [1] Patiño et al. (2025). "Unlocking On-Policy Distillation for Any Model Family".
>
> [2] Zheng et al. (2023). "Judging LLM-as-a-Judge with MT-Bench and Chatbot Arena”.
>
> [3] Bai et al. (2022). “Constitutional AI: Harmlessness from AI Feedback”.

---

> > ### Author Rebuttal · Reviewer_1UMj · 2026-04-01
> >
> > First, thank you for your response, I have updated the score. The rebuttal solve the most important problem about the generalization.
> >
> > For the third question, I still have some additional questions.
> >
> > Since the experiments are done by thinker, it is hard to analysis the memory cost. But I think it is more direct to show the money/credit used for the training? I guess deploying a larger model as teacher introduce additional cost?(I think it is importand to compare the training cost of different methods)

---

> > > ### Author Response · Authors · 2026-04-02
> > >
> > > > Since the experiments are done by thinker, it is hard to analysis the memory cost. But I think it is more direct to show the money/credit used for the training? I guess deploying a larger model as teacher introduce additional cost?(I think it is importand to compare the training cost of different methods)
> > > >
> > >
> > > Thank you for raising the score and your question. Similar to the run time analysis above, **the training cost between multi-turn GRPO and our methods are comparable.** Using the same example as above, on knights and knaves, multi-turn GRPO costs around USD 750 (with USD 600 for the Llama3.1-8B-Instruct model and USD 150 for the Qwen3-235B-A22B-Instruct model), while both of our methods cost around USD 900 (with USD 750 for the Llama3.1-8B-Instruct model and USD 150 for the Qwen3-235B-A22B-Instruct model). Note that again the additional training cost is from more training tokens for the student model instead of the big model. We summarize the comparison below:
> > >
> > > |  | Total | Llama3.1-8B-Instruct | Qwen3-235B-A22B-Instruct |
> > > | --- | --- | --- | --- |
> > > | Multi-turn GRPO | USD 750 | USD 600 | USD 150 |
> > > | Ours | USD 900 | USD 750 | USD 150 |
> > >
> > > To see why the costs are comparable, we want to emphasize again that **the bigger model is not used as a teacher, but instead used to simulate a feedback provider, as a part of the environment,** so the cost of the larger model is unavoidable in this setting, independent from the algorithm. That said, although in our controlled experiment we have to employ such bigger model to simulate the feedback provider, in practice these feedback providers are either human interacting with the LLMs or interpreters, so they automatically provide text feedback in large scale with no additional training cost.
> > >
> > > We hope this addresses the reviewer's final concern. If so, we would be grateful if the reviewer could consider revisiting their evaluation in light of the resolved concerns.

---

### Official Review · Reviewer_q1hQ · 2026-03-12

**Soundness:** 3
**Presentation:** 3
**Significance:** 3
**Originality:** 3
**Overall Recommendation:** 4
**Confidence:** 2

**Summary:**

This paper studies how LLMs can learn effectively from natural language feedback during post-training. It identifies a mismatch between multi-turn RL objectives used during training and the single-turn performance objective. The paper also shows that improving multi-turn reward does not automatically translate into better first-turn responses. To address this, the paper proposes **Self Distillation**, which uses improved second-turn performances, generated after receiving feedback, to directly train the first-turn policy. The method includes importance weightaged estimator and introduces a first-turn reward baseline to prevent gradient collapse when the second-turn performance becomes high. They also introduce **Feedback modeling**, which trains the model to predict textual feedback itself. Empirically, the paper evaluates the proposed methods across reasoning and code generation tasks. Results show **Self Distillation** improves performance over reward-only RL baselines, and that the feedback modeling further enhances performance through self refinement.

**Compliance With Llm Reviewing Policy:**

Affirmed.

**Final Justification:**

My two concerns were both addressed. However, the evaluation scope remains somewhat narrow (two tasks, constrained horizons), and I maintain my score of 4 (Weak Accept). The rebuttal reinforced my prior assessment rather than warranting a score change.

**Key Questions For Authors:**

NA

**Limitations:**

yes

**Strengths And Weaknesses:**

**Strengths**:
The distinction between multi-turn and single-turn objectives is clearly formalized. The unbiased importance weighted estimator is well-motivated. The theoretical analysis of Feedback modeling is another strength. Although the results rely on a local linearized assumption, the claims provide a plausible explanation for why textual feedback offers richer supervision than scalar rewards. Empirical results are solid and support the claim. Comparison to GRPO is appropriate.

**Weaknesses**:
The experiments assume relatively high-quality feedback often produced by a judge model. The robustness of the method to weak feedback is not evaluated. Test time scaling is somewhat limited. Most results focus on two-turn refinement, it would be useful to know whether gains continue or become unstable over longer refinements.

---

> ### Author Rebuttal · Authors · 2026-03-31
>
> We thank the reviewer for the thoughtful review. We are glad the reviewer found the single-turn vs. multi-turn formalization clear, the importance-weighted estimator well-motivated, and the theoretical analysis of Feedback Modeling a strength. We address both concerns below.
>
> >*The experiments assume relatively high-quality feedback often produced by a judge model. The robustness of the method to weak feedback is not evaluated.*
>
> We have added experiments using a smaller, less capable model as the judge, simulating the kind of noisy, imperfect feedback one would expect from non-expert human annotators or lightweight automated graders in practice. We perform experiments on the knights and knaves environment, keep the same learner model, and test a few different judge models and record the results below
>
> | Judge Model  | Self Distillation | Feedback Modeling | GRPO  |
> |-|-|-|-|
> | Qwen3-235B-A22B-Instruct (in the paper) | 0.802  | 0.880 | 0.352 |
> | Qwen3-30B-A3B-Instruct | 0.661 | 0.750 | - |
> | Qwen3-4B-Instruct | 0.609 | 0.758 | - |
>
> We observe that **both of our methods (especially feedback modeling) are robust to the weaker feedback providers** (we use the model from the same model family to show a clean hierarchy), as the performance still greatly improves upon GRPO even with a much weaker model to simulate the feedback provider.
>
> >*Test time scaling is somewhat limited. Most results focus on two-turn refinement, it would be useful to know whether gains continue or become unstable over longer refinements.*
>
> While our main experiments focus on the two-turn setting (where the gap between multi-turn RL and single-turn performance is most clearly isolated), we have added three-turn training (which is the longest horizon under our experiment constraints) and evaluation results:
>
> |  | 2-turn Self Distillation | 2-turn Feedback Modeling | 3-turn Self Distillation | 3-turn Feedback Modeling |
> |-|-|-|-|-|
> | Knights and Knaves | 0.802  | 0.880 | 0.835 | 0.871 |
> | Math500 (DAPO) | 0.548 | 0.567 | 0.568 | 0.577 |
>
> Concretely, in the three turn setting, for self-distillation, we use the same self-distillation objective to distill both $\pi(y_2 \mid x_2)$ and $\pi(y_1 \mid x_1)$ into $\pi(\cdot \mid x_0)$ at the same time, and the extension of feedback modeling is more obvious. We observe improvement on self-improvement on both reasoning tasks and math, and feedback modeling on math as well. We do not observe improvement on feedback modeling on reasoning as the original results already achieve very high accuracy. We also want to document an interesting observation that during the three-turn setting, only distilling the final round teacher $\pi(y_2 \mid x_2)$ induces instability in the training, which we think is worth future investigation on the best style of progressive distillation.
>
> We note that the test-time scaling results in Section 6.4 (Figure 4) also provide some evidence on this question from the inference side: Feedback Modeling enables multiple rounds of self-critique at test time. The new 3-turn training results complement this by showing the behavior on the training side as well.

---

> > ### Author Rebuttal · Reviewer_q1hQ · 2026-03-31
> >
> > Thank you for the rebuttal. My main concerns were the robustness of the method to weaker/noisier feedback and whether the gains extend beyond the two-turn setting. The rebuttal addresses both points satisfactorily.

---

> > > ### Author Response · Authors · 2026-04-01
> > >
> > > We thank the reviewer for confirming that both concerns have been fully resolved. We kindly note that the acknowledgement option selected also suggests considering a score adjustment accordingly, and would be grateful if the reviewer could revisit their score in light of the resolved concerns. We of course fully respect the reviewer's final judgment.

---

### Official Review · Reviewer_XNey · 2026-03-14

**Soundness:** 3
**Presentation:** 4
**Significance:** 3
**Originality:** 4
**Overall Recommendation:** 5
**Confidence:** 3

**Summary:**

Motivated by the limited information in reward- or preference-based post-training and scalability challenges of demonstration-based distillation, the manuscript proposes two methods for leveraging natural language feedback. Prior work has explored multi-turn strategies, where models update their answers based on feedback received after earlier turns. While this improves performance when feedback is available, it typically does not improve first-turn performance in absence of detailed feedback. One proposed method is based on self-distillation via an unbiased estimate of the advantage of correcting the initial prediction by instead providing the updated answer generated after observing feedback. Several stability and efficiency measures are explored, with a baseline of average first-turn rewards providing both more stable convergence and improved final performance compared to measures like clipping importance ratios. The other proposed method is based on feedback modeling, where instead of explicitly correcting predictions, the model is asked to predict the feedback it would receive. This provides a dense learning signal that helps with representation learning, as it targets relevant directions in parameters space that are often missed by the sparse reward signal. Experimental evaluation shows that both methods outperform strong baselines by a considerable margin. Extensive ablation studies show the impact of several design choices. An experiment on test-time scaling with self-critiquing shows that the proposed feedback modeling auxiliary task also adds significant improvements at test time.

**Compliance With Llm Reviewing Policy:**

Affirmed.

**Final Justification:**

My initial evaluation was already very positive and the additional insights provided in the rebuttal reinforced my assessment.

**Key Questions For Authors:**

1. I know it is a general problem with LLM experimentation due to the high computational cost, but I expect there to be significant variance in post-training. Can you comment on variability between runs, how hyperparameters were selected for each model (including baselines)?
2. Could you comment on why you specifically chose Qwen3-235B-A22B-Instruct-2507 to simulate feedback provider and Llama-3.1-8B-Instruct as the main model?

**Limitations:**

yes

**Strengths And Weaknesses:**

## Soundness
The motivation is sound and the theoretical treatment is extensive and well organized. While I did not thoroughly verify every proof, I did not find any mistakes. The theoretical analysis matches the trends observed in empirical results. The only lack in rigour, which this work shares with most LLM papers due to the prohibitive computational cost, is that there is no estimation of the multi-run variability in performance.

## Presentation
The manuscript is well organized and seems polished. I only have a few minor suggestions:
* “Intuition that reward-only RL provides near-rank-1 updates”: I suggest adding a citation here
* I would format the best result in each benchmark in bold (Table 1).
* A short comment on how hyperparameters were chosen would be useful.

## Significance
The proposed methods are addressing the critical question of how to incorporate rich natural language feedback that is only available during RL post-training to improve first-turn performance. Prior methods largely focused on improvements in multi-turn settings, i.e., evaluating how models improve their response after observing feedback on their  initial attempt. If these promising results generalize across more benchmarks, this could be a very impactful contribution.

## Originality
The work proposes to aim for better first-turn performance even when learning from feedback. To my knowledge the two presented methods are novel and show a promising avenue for future research.

---

> ### Author Rebuttal · Authors · 2026-03-31
>
> We thank the reviewer for the thoughtful review. We are glad the reviewer found that the motivation is sound, the theoretical treatment is extensive, the manuscript is well-organized, that our contribution could be impactful, and that this shows a promising avenue for future research. We address each concern below.
> >*I know it is a general problem with LLM experimentation due to the high computational cost, but I expect there to be significant variance in post-training. Can you comment on variability between runs, how hyperparameters were selected for each model (including baselines)?*
>
> We agree it is a general problem with LLM experimentation and we did not have the resources to run all our experiments with multiple seeds. We did run two settings below (one RLTF-FM experiment and one RLTF-SD experiment) with 3 seeds and report the mean and standard deviation in the table below. In these settings, we found the variance to be low and the results to be generally consistent between runs.
>
> |  | Score |
> |-|-|
> | RLTF-FM on Knights and Knaves | 0.828 ± 0.041 |
> | RLTF-SD on Binary Matrix | 0.970 ± 0.005 |
>
> Regarding hyperparameter selection, we mostly used the default settings in tinker for RL baselines. For the auxiliary loss coefficients for our method, we performed a small sweep over knights and knaves and DAPO and selected the best performing ones. However, we only perform the sweep over very few choices over one seed, so we could not claim that our method is operating under the best hyperparameter so far.
>
> >*"Could you comment on why you specifically chose Qwen3-235B-A22B-Instruct-2507 to simulate feedback provider and Llama-3.1-8B-Instruct as the main model?*
>
> We chose these models primarily for practical reasons based on what was available on tinker and relatively cost-efficient. For choosing Llama-3.1-8B-Instruct as the main model, we wanted to use a model with reasonable instruction-following capabilities so that it can incorporate feedback. We chose Qwen3-235B-A22B-Instruct because it was one of the largest models so it is capable of simulating a reliable feedback provider.
>
> We have added experiments using a smaller, less capable model as the judge, simulating the kind of noisy, imperfect feedback one would expect from non-expert human annotators or lightweight automated graders in practice. We perform experiments on the knights and knaves environment, keep the same learner model, and test a few different judge models and record the results below
> | Judge Model | Self Distillation | Feedback Modeling | GRPO  |
> |-|-|-|-|
> | Qwen3-235B-A22B-Instruct (in the paper) | 0.802 | 0.880 | 0.352 |
> | Qwen3-30B-A3B-Instruct | 0.661 | 0.750 | - |
> | Qwen3-4B-Instruct | 0.609  | 0.758 | - |
>
> We observe that **both of our methods (especially feedback modeling) are robust to the weaker feedback providers** (we use the model from the same model family to show a clean hierarchy), as the performance still greatly improves upon GRPO even with a much weaker model to simulate the feedback provider.
>
> >*"”Intuition that reward-only RL provides near-rank-1 updates”: I suggest adding a citation here… I would format the best result in each benchmark in bold (Table 1).*
>
> Thanks for the suggestions. We will add a citation to [1] and update the formatting for Table 1.
>
> [1] Schulman, John and Thinking Machines Lab, "LoRA Without Regret"

---

> > ### Author Rebuttal · Reviewer_XNey · 2026-04-04
> >
> > I thank the authors for the careful and thorough rebuttal. It reinforced my positive impression of the manuscript.

---

> > > ### Author Response · Authors · 2026-04-06
> > >
> > > We are glad to see that our rebuttal addressed the reviewer's questions. We appreciate your support of our work.

---

### Decision · Program_Chairs · 2026-04-30

**Decision:**

Accept (regular)

**Comment:**

This paper studies LLM learning using text feedback which is available at train time but not inference. This problem has been studied before (e.g., Cheng et al. 2023) and its importance stems from the fact that common alternatives like RL learn from scalar reward which is weak, and imitation learning which is expensive and costly. The paper proposes two approaches: one of which is self-distillation where the model mimics its own output conditioned on the feedback, and the second which learns to predict feedback as an auxiliary tasks. This approach seems intuitive and authors have evaluated on a range of tasks and provided additional theoretical analysis.

All reviewers lean towards acceptance. They overall found the approach well-motivated, clearly-written and generally found the results to be strong. Specially, given that text feedback can lead to more sample-efficient learning and improve LLM agents, I think this paper can make a good impact. Therefore, I recommend acceptance.